



# Future evolution of aerosols and implications for climate change in the Euro-Mediterranean region

Thomas Drugé[1], Pierre Nabat[1], Marc Mallet[1], and Samuel Somot[1]

[1]CNRM, Université de Toulouse, Météo-France, CNRS, Toulouse, France

**Correspondence:** T. Drugé (thomas.druge@meteo.fr)

**Abstract.** This study investigates, through regional climate modelling, the surface mass concentration and AOD (Aerosol Optical Depth) evolution of the various (anthropogenic and natural) aerosols over the Euro-Mediterranean region between the end of the $20^{th}$ century and the mid-$21^{st}$ century. The direct aerosol radiative forcing (DRF) as well as the future Euro-Mediterranean climate sensitivity to aerosols have been also analysed. Different regional climate simulations were carried out with the CNRM-

ALADIN63 regional climate model, driven by the global CNRM-ESM2-1 Earth System Model (used in CMIP6) and coupled to the TACTIC (Tropospheric Aerosols for ClimaTe In CNRM) interactive aerosol scheme. These simulations follow several future scenarios called Shared Socioeconomic Pathways (SSP 1-1.9, SSP 3-7.0 and SSP 5-8.5), which have been chosen to analyse a wide range of possible future scenarios in terms of aerosol or particles precursors emissions. Between the historical and the future period, results show a total AOD decrease between 30 and 40% over Europe for the three scenarios mainly

due to the sulfate AOD decrease (between -85 and -93%), that is partly offset by the nitrate and ammonium particles AOD increase (between +90 and +120%). According to these three scenarios, nitrate aerosols become the largest contributor to the total AOD during the future period over Europe, with a contribution between 43.5 and 47.5%. Concerning natural aerosols, their contribution to the total AOD increases slightly between the two periods. The different evolution of aerosols therefore impacts their DRF, with a significant sulfate DRF decrease by 2.6 W m$^{-2}$ and a moderate nitrate and ammonium DRF increase

by 1.4 W m$^{-2}$, on average according to the three scenarios over Europe. These changes, which are similar under the different scenarios, explain about 65% of the annual shortwave radiation change but also about 6% (in annual average) of the warming expected over Europe by the middle of the century. This study shows, with the SSP 5-8.5, that the extra-warming attributable to the anthropogenic aerosols evolution over Central Europe and the Iberian Peninsula during the summer period is due to "aerosol-radiation" as well as "aerosol-cloud" interactions processes. The extra-warming of about 0.2°C over Central Europe

is explained by a surface radiation increase of 5.8 W m$^{-2}$ over this region, due to both a surface aerosol DRF decrease of 4.4 W m$^{-2}$ and cloud optical depth (COD) decrease of 1.3. In parallel, the simulated extra-warming of 0.2°C observed over the Iberian Peninsula is due, as for it, to a COD decrease of 1.3 but also to an atmospheric dynamics change leading to a cloud cover decrease of about 2% and a drier air in the lower layers, signature of the semi-direct forcing. This study thus highlights the necessity of taking into account the evolution of aerosols in future regional climate simulations.



# 1 Introduction

Due to their optical and microphysical properties, atmospheric aerosols are known to have an essential role in the regional and global climate system. Indeed, they are able to modify the entire energy balance as well as cloud properties and the hydrological cycle (Forster et al., 2007; Tang et al., 2018). The Earth's radiative budget is modified by aerosol–radiation interactions (ari), which corresponds to what is usually referred to as the direct and semi-direct aerosol effects, and also by aerosol–cloud interactions (aci) that include what has earlier been referred to as the indirect aerosol effect (Boucher et al., 2013). The direct radiative effect consists in the absorption and scattering of the incident radiation (Mitchell, 1971; Coakley Jr et al., 1983). The absorption of solar radiation by aerosols can also affect the atmospheric thermodynamic properties and dynamics as well as the evaporation of clouds (semi-direct effect, Hansen et al. 1997; Allen and Sherwood 2010). Finally, due to their role as condensation nuclei, aerosols can also modify the microphysical and radiative cloud properties, which notably have consequences on cloud albedo and lifetime (indirect effect, Twomey 1977; Albrecht 1989; Lohmann and Feichter 2005). Over the past few years, efforts have been done in order to quantify the magnitude of the different aerosol radiative effects on the radiative budget (Boucher et al., 2013; Myhre et al., 2013; Stevens, 2015; Allen et al., 2020) but their quantification still shows large uncertainties (Myhre et al., 2020; Bellouin et al., 2020).

The Mediterranean region is characterized by important aerosol loads, composed of both natural (mineral dust and sea-salt) and anthropogenic (sulfate, nitrate, ammonium, black and organic carbon) particles (Lelieveld et al., 2002; Nabat et al., 2013). These aerosols come from different sources such as the Sahara desert, but also from industries, European cities, forest fires and even the Mediterranean Sea itself (Lelieveld et al., 2002). This important variety in aerosol sources leads to complex physico-chemical and optical aerosol properties over the basin, making the Euro-Mediterranean region a particularly interesting area for studying aerosol-climate interactions (Basart et al., 2009; Nabat et al., 2015).

Besides, given the complex Euro-Mediterranean climate, highly dependant on orography and regional winds, the fluxes exchange between the Mediterranean Sea and the atmosphere, the significant contrasts of the surface albedo and the high spatial and temporal variability of aerosols, regional climate models with relatively fine resolution are therefore essential to investigate the aerosol-climate interactions over this region (Gibelin and Déqué, 2003; Gao et al., 2006; Giorgi and Lionello, 2008; Herrmann et al., 2011). Until recently, few studies have addressed the aerosols impact on the Euro-Mediterranean climate using a regional climate model over a multi-year period. Nevertheless, some studies have shown that aerosols have a strong impact on surface radiation, temperature and precipitation in this region (Zubler et al., 2011; Zanis et al., 2012; Spyrou et al., 2013; Nabat et al., 2015; Da Silva et al., 2018; Boé et al., 2020). Indeed, Nabat et al. (2015) showed, thanks to the CNRM-Regional Climate System Model (RCSM) Ocean-Atmosphere coupled regional modelling system, which integrated a realistic aerosol climatology over the period 2003-2009 (Nabat et al., 2013), a surface shortwave direct aerosol radiative forcing (DRF) of the order of -20 W m$^{-2}$ over the Mediterranean Sea and North Africa and of the order of -15 W m$^{-2}$ over Europe. Other studies, such as that of Papadimas et al. (2012), have shown similar values over the Mediterranean basin with a mean aerosol DRF at the surface of about -16.5 W m$^{-2}$. Over Europe and the Mediterranean Sea, the cooling effect shown by Nabat et al. (2015) at the surface is partially offset by the semi-direct radiative effect, consequence of a change in cloud cover and





atmospheric circulation. At the surface, the net radiative forcing (direct + semi-direct aerosol effect) is therefore negative over Europe, the Mediterranean Sea and North Africa (-16.2, -11.7 and -21.0 W m$^{-2}$ respectively) and consequently leads to a surface cooling of about 0.5°C over the Mediterranean basin. Using a RegCM3 simulation, Zanis et al. (2012) showed in their study that the anthropogenic aerosols induced small changes in the yearly averaged surface temperature over Europe during

the period 1996-2007 with a maximum temperature drop in the order of 0.2°C over the Balkan Peninsula. At the top of the atmosphere (TOA), the direct aerosol radiative forcing is more contrasted, with values that can be positive over North Africa (especially in summer) and negative values over the Mediterranean Sea and Europe, due to surface albedo differences. Over a longer period of time, Nabat et al. (2014) have also shown the important role of aerosols in the climate trends observed in recent decades. In particular, they have shown that the sulfate aerosol decrease since the 1980s over Europe and the Mediterranean

Sea has largely contributed to the incident solar radiation increase at the surface (81 ± 16%) as well as to the warming (23 ± 5%) over the Mediterranean region. This phenomenon has also been studied over Europe by Zubler et al. (2011) with the COSMO-CLM version over the period 1970-2000. In particular, they showed a strong brightening of up to 10 W m$^{-2}$ in the annual mean over this period for mid-Europe.

The climate of the Mediterranean region, alternating between hot and dry summers and mild and wet winters, is highly

reactive to current global climate change. The Mediterranean area is even considered as a climate change "hotspot" (Giorgi, 2006). Indeed, Jacob et al. (2014) have showed, based on a set of 7 high-resolution regional models from the CORDEX project, a strong temperature increase at the end of the century (2071-2100) compared to the period 1971-2000, that is of the order of 1 to 4.5°C with the representative concentration pathways (RCP) 4.5 and of the order of 2.5 to 5.5°C with the RCP 8.5. Other studies from the ENSEMBLES project have also predicted a temperature increase of the same order of magnitude with the A1B

scenario (Hewitt, 2004; van der Linden and Mitchell, 2009). These various studies have also shown a significant precipitation increase of at least 25% over a large part of Central and Northern Europe, as well as a precipitation decrease around the Mediterranean Sea. Other studies such as those of Giorgi and Lionello (2008) or of Thiébault and Moatti (2016) confirm these results. Given this Mediterranean climate sensitivity to climate change, but also the important aerosol radiative effect on the past and present climate of this region mentioned above, it is necessary to take into account the aerosol evolution to study the

impact of these particles on the future Euro-Mediterranean climate. Boé et al. (2020) and Gutiérrez et al. (2020) have also recently highlighted the need to take into account the anthropogenic aerosol evolution in order to study the future evolution of climate over Europe, and in particular the solar radiation changes. For the time being, a few studies have only addressed the variability of the surface fine particles mass concentration, in particular the PM$_{10}$ and PM$_{2.5}$ using air quality prediction models. These studies have shown a decline in PM$_{10}$ over Europe in future (Lacressonnière et al., 2014; Markakis et al., 2014;

Lacressonnière et al., 2017; Cholakian et al., 2019). The intensity of this decrease depends on the period that is taken into account and on the inputs used. Cholakian et al. (2019) showed in particular a PM$_{10}$ concentration decrease, in the 2050s, of about 12% over Europe for the RCP 4.5.

Nevertheless, there are still few studies dedicated to investigate the impact of anthropogenic and natural aerosols on the future Euro-Mediterranean climate. Moreover, to our knowledge, this question has never been addressed by using a regional

climate model with interactive aerosols providing a realistic description of the different particles and allowing the study of





their impact on the climate. In that context, the first aim of this study is to investigate the evolution of the different (natural and anthropogenic) aerosol loads (mass surface concentration and aerosol optical depth) in future, using the CNRM-ALADIN63 regional climate model and its TACTIC aerosol scheme (Nabat et al., 2020). In addition, the use of the recent anthropogenic emission dataset provided for the sixth phase of the Coupled Model Intercomparison Project (CMIP6) allows to have more

realistic emissions than those of previous studies. In a second time, this regional modelling tool will serve to address the evolution of direct radiative forcing and the impact of aerosol evolution on the future Euro-Mediterranean climate change. After a description of the CNRM-ALADIN63 regional model and its aerosol scheme in Sect. 2, the evolution of the different aerosols as well as their radiative forcing will be presented in Sect. 3. The future Euro-Mediterranean climate sensitivity to aerosols will be discussed in Sect. 4 before the concluding remarks in Sect. 5.

## 2   Model and experiments

### 2.1   The CNRM-ALADIN63 regional climate model

The current work has been carried out with the CNRM-ALADIN63 regional climate model (Nabat et al., 2020), that is developed at the CNRM (National Centre for Meteorological Research). It includes an interactive aerosol scheme described thereafter. It includes also the SURFEX land surface module (Masson et al., 2013) with the ISBA scheme (Noilhan and Plan-

15 ton, 1989) that models water and energy exchanges at the interface between the plant, the soil and the atmosphere. SURFEX will then provide the mean values of different data such as the ascending radiation or the surface albedo which are necessary for the CNRM-ALADIN63 model radiative scheme. Lastly, land surface hydrology and river flow are simulated by the TRIP model (Oki and Sud, 1998) according to the evaporation-precipitation balance provided by ISBA.

CNRM-ALADIN63 is a bi-spectral, hydrostatic limited-area regional climate model with a semi-Lagrangian advection and

20 a semi-implicit scheme. The configuration of CNRM-ALADIN63 used here over the Euro-Mediterranean region has a 50 km horizontal resolution and 91 vertical levels. Contrary to the globe, the domain is not periodic, so an extension zone used only for Fourier transforms has been added with the aim of achieving the bi-periodization. CNRM-ALADIN63 model includes the Fouquart and Morcrette shortwave radiation scheme (FMR, Fouquart and Bonnel 1980; Morcrette et al. 2008) with six spectral bands, and the RRTM (Rapid Radiative Transfer Model) longwave radiation scheme (Mlawer et al., 1997).

The Euro-Mediterranean domain chosen for this study, presented in Figure 1, includes the official domain of the Med-CORDEX initiative (Ruti et al., 2016) as well as an extension to take into account the two main dust sources (the Sahara and the largest part of the Arabian Peninsula). This domain represents 128 x 180 points, including a bi-periodization of 11 points (north and east) and a classical relaxation zone of 8 points (on each side). The Europe and the Mediterranean Sea regions are highlighted in Figure 1 for the needs of this study.



## 2.2 The TACTIC aerosol scheme

CNRM-ALADIN63 incorporates an interactive aerosol scheme named TACTIC (Tropospheric Aerosols for ClimaTe In CNRM, Michou et al. 2015; Nabat et al. 2015; Watson et al. 2018; Drugé et al. 2019; Michou et al. 2020; Nabat et al. 2020), which is originally based upon the GEMS/MACC aerosol module of the ECMWF operational forecast model (Morcrette et al., 2009).

The version of this aerosol scheme used in this study is described in detail in Nabat et al. (2020).

Initially, this aerosol scheme takes into account five aerosol types (desert dust, sea salt, black carbon, organic matter and sulfate), and recently nitrate and ammonium particles have been added as presented in Drugé et al. (2019). The TACTIC aerosol scheme makes it possible to simulate the aerosol life cycle (emission, transport and deposition), considering the different aerosols as prognostic variables, and taking into account their direct and semi-direct radiative effects. The first indirect aerosol

effect (cloud albedo) is also represented for sulfate, organic matter and sea salt aerosols. On the other hand, the second indirect effect (interactions between aerosols and cloud microphysics) is not included in this aerosol scheme. Finally, secondary organic aerosols (SOA) are taken into account through the climatology of Dentener et al. (2006), but their formation is not explicitly included in the TACTIC aerosol scheme.

To represent the particle size spectrum, the TACTIC aerosol scheme includes 16 prognostic variables using several size bins

for some specific species. In detail, three size bins are used for dust aerosols (0.01 to 1.0, 1.0 to 2.5 and 2.5 to 20 μm) and for primary sea-salt (0.01 to 1.0, 1.0 to 10.0 and 10.0 to 100.0 μm). Organic matter and black carbon aerosols are both separated into two different bins (hydrophilic and hydrophobic particles). Only one bin is used for sulfate particles and another is used for its precursors, notably sulphur dioxide ($SO_2$). Recently, nitrate aerosols have been included in the TACTIC aerosol scheme (Drugé et al., 2019). This specific tracer is divided into two different bins (for fine and coarse modes). The last two bins are

used for ammonium and ammonia tracers. It is important to note that one of the ammonium and nitrate aerosols precursors, the nitric acid $HNO_3$, is implemented in the model as a constant monthly climatology based on the CAMS reanalysis (Inness et al., 2019). This nitrate-ammonium module is described in detail in Drugé et al. (2019). The different primary aerosol mass concentrations can be dynamically calculated on-line as a function of surface wind and soil characteristics (dust, primary sea-salt) or based on external emission datasets from anthropogenic and/or biomass burning particles (black carbon, organic matter,

sulfate, nitrate and ammonium). After being emitted into the atmosphere, all aerosol species are then transported and submitted to the dry and wet (in and below clouds) deposition. Finally, it is important to note that, in the present study, the defined ALADIN domain is supposed to be large enough to include all aerosol sources affecting the Euro-Mediterranean region and the long-range transport of particles is not included in the lateral boundary forcing.

The different radiative properties of each aerosol species, and notably those used in input for the radiative transfer scheme

(the extinction coefficient, asymmetry parameter and single scattering albedo) are set for each aerosol type following Nabat et al. (2013) and Drugé et al. (2019). These different aerosol optical properties are pre-calculated using a Mie code using the hypothesis of aerosol sphericity (Ackerman and Toon, 1981) and are dependant on relative humidity, with the exception of mineral dust and hydrophobic black and organic carbon particles.





The different simplifications presented previously, such as the limited number of bins used for anthropogenic aerosols, the hypothesis of the external mixing state between the different species or the fact that the aerosol optical properties are not calculated on-line in the model, are both necessary to keep the regional model in a reasonable computation cost, which is in this configuration well adapted to perform multi-decadal simulations.

## 2.3 Simulations

The TACTIC aerosol scheme presented previously allows us to study in the following section the variation of aerosols between the past (from 1971 to 2000) and the future (from 2021 to 2050) period. The future period has been selected in the near future because, unlike greenhouse gases, the most important aerosol change is up to the middle of the century (see Figures 2 and 3). Moreover, the near future horizon period is most suitable to help public decision-makers. For this purpose, a first simulation over the past period (1971-2000) and three simulations over the future period (2021-2050) were carried out. These future ALADIN simulations are based on three Shared Socio-economic Pathways (SSPs, O'Neill et al., 2017), namely SSP 1-1.9, SSP 3-7.0 and SSP 5-8.5. The choice of these scenarios will be explained below (Sect. 3.1). These four simulations will be named Hist, SSP119, SSP370 and SSP585 respectively. All these different simulations use the CMIP6 dataset for the emission of the different aerosols or aerosol precursors. The historical dataset is based on two main sources, namely the anthropogenic emissions (Hoesly et al., 2018) and the biomass burning emissions (Van Marle et al., 2017). The anthropogenic emissions of the different aerosols and their precursors are provided from the main sectors of activity (residential, agriculture, energy, industrial, air, land and sea transport, waste and solvents), for each country and for each grid point at 0.5 degree resolution, while the emissions from biomass fires (natural and anthropogenic) are provided over the period 1750-2014. Future CMIP6 emissions are provided at the same format, derived from the different SSPs (Gidden et al., 2019). Finally, a fifth simulation, named SSP585cst, which will be used in Sect. 4 of this study was also carried out with the SSP 5-8.5. This simulation is similar to the SSP585 simulation but with constant aerosol and aerosol precursors emissions that correspond to the average historical emissions over the 1971-2000 period. As a reminder, these five simulations were carried out, at 50 km resolution, with the CNRM-ALADIN63 model including the TACTIC aerosol scheme and being uncoupled with the ocean. No specific spinup was used for these simulations but the restart is that of the end of the simulations presented in Drugé et al. (2019). Atmospheric lateral boundary conditions, sea surface temperature, sea ice cover and ozone concentrations come from historical and respective scenario simulations carried out with the global Earth system model CNRM-ESM2-1 (Séférian et al., 2019) presented below. Land use changes are not considered in this study. The historical evolution of the different greenhouse gases (GHGs) is included in these simulations following the yearly global averages of Meinshausen et al. (2017). Besides, following Matthes et al. (2017), the total solar irradiance forcing is also taken into account with yearly averages. Finally, the stratospheric aerosol radiative forcing, including the contribution of the main historical volcanic eruptions such as Mt. Pinatubo in 1991, is included through the Thomason et al. (2018) data set, providing stratospheric AOD at 550 nm. Table 1 summarises the characteristics of the different simulations used in this study.

All these simulations are driven by the CNRM-ESM2-1 Earth system model (Séférian et al., 2019), which has a horizontal resolution of about 150 km. This global climate model, which contributes to the sixth phase of the Coupled Model Intercom-



parison Project (CMIP6, Eyring et al. 2016), is developed by the CNRM and CERFACS (European Center for Research and Advanced Training in Scientific Computing) modelling groups. The physical core of this climate model is the global coupled ocean-atmosphere model CNRM-CM, whose version 6 is used here (Voldoire et al., 2019). CNRM-ESM2-1 is mainly dedicated to the realization of climate scenarios and is composed of different models such as the ARPEGE-Climate model

(Roehrig et al., 2020) for the atmosphere, NEMO for the ocean (Madec et al., 2017), and REPROBUS (Lefevre et al., 1994) for the chemistry. Like the CNRM-ALADIN63 model, this global model also includes the TACTIC prognostic aerosol scheme, including all the aerosols present in TACTIC with the exception of nitrate and ammonium particles. The CNRM-ESM2-1 model also integrates the SURFEX continental surface modelling platform (Masson et al., 2013) with the ISBA-CTRIP land surface model (Decharme et al., 2019) and a lake model named Flake (http://www.flake.igb-berlin.de/, Le Moigne et al., 2016).

Finally, the CNRM-ESM2-1 model also integrates a sea ice scheme (GELATO, Mélia 2002) and a marine biogeochemistry module (PISCES, Aumont and Bopp 2006). All these CNRM-ESM2-1 model components are presented in detail in Séférian et al. (2019). It is worth mentioning that CNRM-ALADIN63 and CNRM-ESM2-1 share the same physic basis (atmosphere, surface, aerosols) and the same climate forcings (solar forcing, GHGs, aerosol emissions, ozone) as far as possible.

## 2.4   Anthropogenic CMIP6 emissions

As mentioned above, the CMIP6 emissions of aerosols or aerosol precursors are provided by Hoesly et al. (2018) and Van Marle et al. (2017) for the historical period (1750-2014), and by Gidden et al. (2019) for the future period (2015-2100) following the various SSPs. The anthropogenic emissions, ranging from 1971 to 2100 over Europe, are presented in Figure 2 for sulphur dioxide, ammonia, black carbon and organic carbon according to the different existing scenarios. Compared to the previous CMIP5 inventory, which has been improved by taking better account of the different emission sectors as well as re-evaluated

emission factors, this CMIP6 dataset presents some differences as showed in Figure 2, in particular for ammonia emissions. Indeed, CMIP6 anthropogenic ammonia emissions show a decrease from the 1990s, whereas the previous CMIP5 inventory showed the opposite trend with an increase in ammonia emissions. Despite this, for other aerosols or aerosol precursors, trends are similar between the two datasets, such as the sulphur dioxide which shows a peak in the early 1980s followed by a significant decrease in both emission inventories. Figure 2 shows a decrease in emissions over the historical period for all

aerosols, particularly from the 1990s. The most significant decrease, by about 80%, concerns sulphur dioxide, a precursor of sulfate aerosols, which had the highest emissions. Indeed, sulphur dioxide emissions were approximately divided by five between 1980 and 2010. Concerning the different scenarios, this figure shows that, with the exception of ammonia, the majority of them predict a decrease in the different emissions in the future. However, in the ammonia case, several scenarios such as the SSP 5-3.4, the SSP 4-6.0 or the SSP 3-7.0 predict higher emissions by 2100 compared to 2010 (+60, +10 and +20%

respectively).

The choice of the three scenarios used in this study gives a contrasted range of values in terms of radiative forcing but also in terms of aerosol or aerosol precursor emissions. In addition, as shown in Figure 3 which presents the $CO_2$ concentration evolution, the SSPs 1-1.9 and 5-8.5 allow us to have the widest possible range of future radiative forcing. Lastly, the SSP 1-1.9 is interesting as it represents one of the few combinations that can be used to comply with the Paris Agreement and thus limit





the global temperature increase to 1.5°C by 2100. The SSP 3-7.0 was chosen as an intermediate scenario between the SSP 1-1.9 and the SSP 5-8.5 and because it is one of the scenario characterized by an ammonia emission increase. These different scenarios thus make it possible to study the evolution of all the aerosols present over the Euro-Mediterranean region thanks to a wide range of possible futures in terms of concentration and optical thickness of the different aerosols.

## 3    Future evolution of aerosols and their radiative forcing

### 3.1    Surface mass Concentration and AOD evolution

The total AOD evolution between the past (1971-2000) and the future (2021-2050) periods can be seen in Figure 4. In more details, Figure 5 presents the aerosol evolution between these two periods in terms of surface mass concentration and AOD for each aerosol type over both Europe (a) and the Mediterranean Sea (b). First of all, Figure 5 indicates that the total aerosol surface mass concentration over the Mediterranean Sea is significantly higher than over Europe, in particular due to the large amount of natural aerosols such as desert dust or primary sea salts over this area. Nevertheless, over the period 1971-2000, the total AOD over Europe is similar to that over the Mediterranean Sea, of the order of 0.2. Concerning the aerosol evolution between the historical period 1971-2000 and the future period 2021-2050, Figures 4 and 5 show a future concentration and total AOD decrease over Europe and the Mediterranean Sea. In terms of AOD, this decrease is equivalent to about -0.07 on annual average according to the SSP 5-8.5 over Europe and -0.02 over the Mediterranean Sea. This total AOD decrease is explained, in part, by the strong decrease in sulfate concentration and AOD that is notable between the past and the future periods in all months of the year. Indeed, over Europe, sulfate AOD decreases from 0.12 on average over the historical period to 0.01 over the future period according to the SSP 5-8.5, i.e. a decrease of 90%. Similarly, over the Mediterranean Sea, this decrease is of the order of 85%. However, this sharp sulfate AOD drop and thus the total AOD decrease is compensated by a nitrate and ammonium AOD increase of about 30 %, as shown in Figure 5. On average over Europe (Figure 5, a), the nitrate AOD increases from 0.03 (0.01 for ammonium) over the period 1971-2000 to 0.06 (0.02 for ammonium) over the period 2021-2050, i.e. a 100% increase again according to the SSP 5-8.5. Over the Mediterranean Sea (Figure 5, b), the AOD of theses particles increases by about 300% for nitrate aerosols and about 230% for ammonium aerosols. Figure 5 shows also that the other aerosols (dust, sea salt, organic carbon and black carbon) do not show strong concentration or AOD changes between the two periods studied, either over Europe or over the Mediterranean Sea.

Finally, Figure 4 also shows a total AOD increase over the western and eastern part of Africa according to the three scenarios studied. This increase is due both to a nitrate and ammonium AOD increase but also to an increase in dust AOD during the summer and in particular during the months of July and August over this region. This result is illustrated in Figure 6 which shows the different aerosol evolution between the period 1971-2000 and the period 2021-2050, for the SSP 5-8.5, over the East African region shown in black. The dust AOD increase during the summer between the two periods is due to a wind speed increase, shown in Figure 7, and therefore higher dust emissions. Figure 7 also shows that this wind increase during the summer present with the CNRM-ALADIN63 model is also found with its forcing model (CNRM-ESM2-1) as well as with the average of the different members carried out with this global model. Concerning the nitrate and ammonium AOD increase



over this region, it is due to an increase in ammonia emissions, but also to an increase in dust emissions and therefore to more calcite available which will then react with nitric acid to form nitrate and ammonium aerosols (Drugé et al., 2019).

The most important concentration and AOD changes between the past and the future period therefore occur over Europe and are mainly due to a strong sulfate concentration and AOD decrease, which is partly offset by the nitrate and ammonium

concentration and AOD increase. Figures 4 and 5 show that these changes are robust to the three scenarios used in this study. Nevertheless, the SSP 3-7.0 shows the smallest total AOD decrease over Europe with a decline of 0.06, compared to 0.07 with the SSP 5-8.5 and 0.08 with the SSP 1-1.9. This is due to a more moderate sulfate AOD decrease and a slight nitrate AOD increase, initially due to higher sulphur dioxide and ammonia emissions with this scenario. Except for these few variations, all results showed a total AOD decrease over Europe for the three scenarios (SSP 1-1.9, SSP 3-7.0 and SSP 5-8.5). These results

are therefore in agreement with the first studies that addressed the fine particles evolution, and in particular $PM_{10}$ and $PM_{2.5}$, which forecast a fine particles decrease over Europe by the end of the century (Markakis et al., 2014; Lacressonnière et al., 2017; Cholakian et al., 2019). These results are also consistent with Boé et al. (2020) and Gutiérrez et al. (2020) studies that predict a decline in total aerosol AOD by the middle of the century.

Following the evolution of these different aerosol types, their contribution to the total AOD will therefore also change

between the past and the future periods. Table 2 indicates the relative contribution of each aerosol type to the future total AOD (2021-2050) over Europe and the Mediterranean Sea under the three different scenarios. This table shows that nitrate particles are the highest contributor to the total AOD over Europe for the future period, with a contribution of about 45% over the period 2021-2050 according to the different SSPs compared to only 15% over the period 1971-2000. The contribution of these aerosols to the total AOD is the highest (47.5%) when using the SSP 1-1.9. These results are consistent with Bellouin et al. (2011) and

Drugé et al. (2019) studies, which show that nitrate particles are or would become the dominant aerosol species over Europe. In addition Drugé et al. (2019) show, over the period 1979-2016, that the ammonium and nitrate concentrations increase over Europe is due to the decrease in sulfate aerosol production which produces more free ammonia in the atmosphere. At the global scale, Papadimas et al. (2012) also showed that nitrate particles will increase from a contribution of 23% to anthropogenic AOD (year 2000) to 56% (year 2090) for the RCP 4.5 scenario. Over the Mediterranean Sea, nitrate aerosols become the

second highest contributor to the total AOD, of about 20% according to the three SSPs, just behind desert dust particles, which contribute of about 40% to the total AOD. In addition to nitrate particles, the ammonium aerosol contribution to the total AOD over Europe also shows a clear increase between the two periods by being multiplied by three, from 5 to 15%. Conversely, sulfate particles, which are the highest contributors (60%) to the total AOD during the period 1971-2000 over Europe would contribute only at 10% to the future total AOD with the SSP 5-8.5 (7% with the SSP 1-1.9 and 13% with the SSP 3-7.0). Over

the Mediterranean Sea, sulfate also shift from the highest contributor (38.5%) to one of the least contributing species to the total AOD except with the SSP 3-7.0 where sulfate particles have a contribution to the total AOD of about 16%, comparable to that of nitrate (19%) or primary sea salt (15%) aerosols. Finally, natural aerosol contribution to the total AOD increases slightly between the two periods both over Europe and the Mediterranean Sea. For information the increase in sea salt emissions over the Mediterranean Sea, particularly in spring and summer, is explained by the increase in sea surface temperature, on which

these aerosols emission directly depends, and not by a change in wind speed.



### 3.2 Evolution of aerosol direct radiative forcing

The evolution and variability of the different aerosol species over the Euro-Mediterranean region, presented above, is mainly dominated by the sulfate mass concentration decrease partially compensated by the nitrate concentration increase. This evolution would consequently impact the variability of the shortwave (SW) direct radiative forcing (DRF) exerted by the anthropogenic particles. Aerosol direct radiative forcing, at the surface and TOA in SW and LW spectral ranges, is diagnosed using a double call (with and without aerosols) to the radiation scheme during the model simulation.

### 3.2.1 Historical aerosol DRF

Before studying the change in the DRF of the different particles between the historical (1971-2000) and the future period (2021-2050), we first analyse the SW DRF exerted by all aerosols over the historical period. The results are presented seasonally both at the surface and at the TOA in Figure 8 and summarised in Table 3. Unless otherwise specified, the different presented direct radiative forcing values are calculated in all-sky conditions. Figure 8 shows that natural and anthropogenic aerosols, by absorbing and scattering solar radiation, cause a surface SW DRF estimated on average of -5.7 W m$^{-2}$ over Europe, -8.7 W m$^{-2}$ over the Mediterranean Sea and -10.6 W m$^{-2}$ over North Africa. In comparison, some studies using multi-annual regional simulations have been carried out. For example, Nabat et al. (2015) showed a surface SW DRF over the period 2003-2009 of the order of -14.7 W m$^{-2}$ over Europe, -20.9 W m$^{-2}$ over the Mediterranean Sea and -19.7 W m$^{-2}$ over North Africa. Differences between this study results and those presented here are largely explained by a different version of the atmospheric model between the two modeling exercises, but also by the non-inclusion of nitrate and ammonium aerosols and the use of an aerosols climatology in the study of Nabat et al. (2015). Another work, using the regional climate model RegCM, which includes an aerosol prognostic scheme for five species (sulfate, organic carbon, black carbon, dust and primary sea salt) showed, over the period 2000-2009, a mean surface DRF close to that simulated here, of the order of -13.6 W m$^{-2}$ over the Mediterranean Sea and -14.9 W m$^{-2}$ over North Africa (Nabat et al., 2012). Benas et al. (2013) have shown for their part a mean surface aerosol DRF between -15 and -30 W m$^{-2}$ in Crete over the period 2000-2010. Finally, the Papadimas et al. (2012) study indicates, on the basis of MODIS satellite data, a mean DRF over the Mediterranean basin of the order of -16.5 W m$^{-2}$ at the surface (period 2000-2007), with a maximum during spring and summer. Table 3 as well as Figure 8 also show that the highest SW DRF simulated at the surface occurs during spring, especially for regions with high dust emissions (-17.3 W m$^{-2}$ over Africa) and over the Mediterranean Sea (-13.1 W m$^{-2}$) near the coasts. The results presented above are therefore fairly consistent with these various studies, both on the intensity and on the spatial and temporal distribution of historical surface SW DRF.

   At the TOA, the aerosol SW DRF is also negative (-3.3 W m$^{-2}$ over Europe and -4.8 W m$^{-2}$ over the Mediterranean Sea on average) with a maximum in summer, except over North Africa where it is slightly positive (+2.8 W m$^{-2}$) due to a high surface albedo over this region and the absorbing properties of mineral dust (single scattering albedo of dust fixed at 0.90 for the 550 nm wavelength in CNRM-ALADIN63). This result is consistent with the study of Bellouin et al. (2011), carried out with a global climate model, which shows a radiative forcing (direct and first indirect effect) of the order of -4 W m$^{-2}$ at the





TOA over Europe for the year 2000. Moreover, over the period 2000-2007, Papadimas et al. (2012) estimated a DRF at the TOA of -2.4 W m$^{-2}$ over the Mediterranean basin as well as a positive DRF over desert areas of +4.1 W m$^{-2}$.

Whether on the surface or at the TOA, the CNRM-ALADIN63 model therefore simulates a historical aerosol SW DRF which is consistent with previous studies. Therefore, this model seems to be valid for studying the future evolution of this aerosol DRF.

### 3.2.2   Future aerosol DRF

Figure 9 presents finally the total SW aerosol DRF difference between the historical (1971-2000) and the future (2021-2050) periods according to the three different scenarios previously presented. This figure shows a similar DRF evolution using the three different scenarios with a significant decrease in absolute terms, at the surface and at the TOA over Europe, in particular during the summer over the Po Valley, the Benelux and Eastern Europe, up to 6 W m$^{-2}$. The total DRF decrease over Europe at the surface is 1.6 W m$^{-2}$ on average with the SSP 5-8.5, 1.2 W m$^{-2}$ with the SSP 3-7.0 and 2.0 W m$^{-2}$ with the SSP 1-1.9. The SSP 1-1.9 shows the largest decrease in total DRF over Europe, which is consistent with the total AOD evolution analysed previously. On the other hand and over North Africa, Figure 9 shows an increase in the surface DRF, in absolute terms, ranging from 0.5 W m$^{-2}$ with the SSP 1-1.9 to 1.4 W m$^{-2}$ with the SSP 3-7.0. At the TOA, the same trends are found but with lower values of the order of 1 W m$^{-2}$ over Europe and 0.2 W m$^{-2}$ over North Africa. These changes in DRF are therefore consistent with the changes in total AOD presented previously. Moreover, these results are also consistent with Shindell et al. (2013), who showed a lower aerosol radiative forcing (direct and indirect effects) in 2030 than in 1980 over Europe with global climate models.

The evolution, between the two periods, of the SW DRF exerted at the surface by each aerosol type, which is diagnosed using several calls to the radiation scheme at each time step, is reported in Figure 10. The same results, at the TOA, are presented in Appendix in Figure A1. Showing the same trends as the results observed at the surface, but with lower values, only the evolution of the SW DRF exerted at the surface by each aerosol type will be discussed here. As showed previously the results clearly indicate, in absolute terms, that the aerosol DRF decreases over Europe between the two periods. This is largely due to sulfate and organic carbon aerosols, which represent a mean DRF decrease at the surface of 2.6 W m$^{-2}$. On the other hand, nitrate and ammonium aerosols partially compensate the sulfate and organic carbon DRF decrease with a contribution of 1.4 W m$^{-2}$ on average for the three SSPs. The evolution of the different aerosol DRF estimated here is consistent with results obtained at the global scale by Hauglustaine et al. (2014). Our results concerning the nitrate DRF are also consistent with other studies carried out at global scale. Indeed, the global nitrate DRF is estimated to be between -0.02 and -0.19 W m$^{-2}$ (Adams et al., 2001; Jacobson, 2001; Liao and Seinfeld, 2005; Bauer et al., 2007; Xu and Penner, 2012; IPCC, 2013; Myhre et al., 2013) and is estimated to be between -0.4 and 1.3 W m$^{-2}$ by the end of the century, becoming the aerosol with the highest cooling effect (Adams et al., 2001; Bellouin et al., 2011). Figure 10 also shows that the natural aerosol DRF remains relatively stable between the two periods. Over the Mediterranean Sea, Figure 10 shows a surface aerosol DRF evolution close to the one simulated over Europe, with a sulfate and organic carbon DRF decrease, of about 2.6 W m$^{-2}$, associated to an nitrate and ammonium DRF increase of about 1.7 W m$^{-2}$. Both over Europe and the Mediterranean Sea, the largest sulfate and organic



carbon DRF decrease occurs with the SSP 1-1.9 (2.8 W m$^{-2}$ on average over Europe and 3.0 W m$^{-2}$ over the Mediterranean Sea). As presented previously, this specific scenario presents the largest concentration decrease of these particles over these two regions. Concerning North Africa, Figure 10 indicates a slight total DRF increase, which is partly due to an increase of nitrate and ammonium DRF (of about 0.6 W m$^{-2}$ for the three scenarios), but also for black carbon aerosols, especially for

SSPs 3-7.0 and 5-8.5 (0.3-0.4 W m$^{-2}$). In addition, this total DRF increase over North Africa is also due to a moderate dust DRF increase, which is about 0.3-0.4 W m$^{-2}$ over this region.

## 4    Future Euro-Mediterranean climate sensitivity to anthropogenic aerosols

The evolution of the surface mass concentration, AOD and DRF of the different aerosols discussed previously are expected to have some significant impacts on the future Euro-Mediterranean climate. As primary natural sea-spray and mineral dust

aerosols are characterized by moderate changes between the historical and the future period, only the anthropogenic aerosols impact will be studied here. Furthermore, as the differences between the scenarios are small, only the SSP 5-8.5 will be discussed in this section for clarity reasons.

### 4.1    Future climate simulated by CNRM-ALADIN63

First of all, Figure 11 presents the changes in surface SW radiation (a), surface temperature (b), precipitation (c) and surface

wind speed (d) between the historical and future periods over the Euro-Mediterranean region for the SSP 5-8.5. This figure (a) shows a sharp surface SW radiation increase over Europe between the two periods (1971-2000 and 2021-2050), that is of the order of 5 W m$^{-2}$ on annual average. Figure 11 (a) indicates also a strong seasonal variability with a higher surface radiation increase in summer (7.5 W m$^{-2}$). Concerning surface temperatures, this figure (b) indicates a significant increase over the Euro-Mediterranean region. On annual average, the increase between the two periods is about 2.2°C over Europe,

1.7°C over the Mediterranean Sea and 1.9°C over Africa. The strongest temperature increase over Europe is simulated during the winter, with an average of 2.5°C and a maximum of 4°C over Eastern Europe. Over Africa, the maximum temperature increase occurs in summer with an average of 2°C. These results are consistent with different studies such as that of Jacob et al. (2014), Boé et al. (2020) or Coppola et al. (In review). This last study shows, for the RCP 8.5 or the SSP 5-8.5, a median warming in winter of 2.5°C for the CMIP5 ensemble and of 3.8°C for CMIP6 over Central Europe for the mid of the century

(2041-2070) relative to 1981-2010. Over the Mediterranean, they have shown a smaller warming between 2°C and 2.5°C. Finally, Coppola et al. (In review) also show a summer warming that is similar to that shown in Figure 11 with a maximum signal over the Mediterranean land regions. Moreover, the ENSEMBLES project results show a temperature increase from 3 to 4.5°C, also more pronounced in winter, with the A1B scenario over this region between 1971-2000 and 2021-2050 (Hewitt, 2004; van der Linden and Mitchell, 2009). Concerning precipitation and surface wind speed, except during winter, Figure 11

(c and d) shows few areas with significant changes. These results therefore present a higher uncertainty than temperatures. Nevertheless, Figure 11 shows an average rainfall decrease of about 5% (2% for surface wind speed) over the Mediterranean Sea between the period 1971-2000 and the period 2021-2050 with a maximum of 10% (5% for surface wind speed) in winter.





Over Europe, and mainly in the northern part of Europe, an increase in precipitation can be observed, with a maximum of 1 mm day$^{-1}$ in the Alps. The only season with a decrease in precipitation throughout Europe is the summer with an average decrease of 0.2 mm day$^{-1}$. Similar results concerning precipitation are observed in different studies (Jacob et al., 2014; Boé et al., 2020; Coppola et al., In review) as well as in the ENSEMBLES project from scenario A1B. Indeed, Coppola et al. (In

review) have shown, for the mid of the century and in winter, a median precipitation increase of 10-12% over Central Europe and a median negative change of 5% over the Mediterranean for the EURO-CORDEX, the CMIP5 and the CMIP6 ensembles. For the summer period, they showed a negative change of precipitation spanning the range from 4 (EURO-CORDEX) to -10% (CMIP6) over Central Europe and from -10 to -20% over the Mediterranean. Finally, compared to these different ensembles, the simulation used in this study is therefore an average simulation, close to the CMIP5 ensemble.

The question of the part of these different changes which can be explained by the evolution of aerosols will now be addressed. For that, Figure 12 shows the role of anthropogenic aerosols, by difference between SSP585 and SSP585cst simulations, in surface radiation (a), surface temperature (b), precipitation (c) and surface wind speed (d) changes previously seen between the historical and the future period. This figure (a) shows that the various anthropogenic aerosols are responsible for about 65% of the radiation increase, on annual average over Europe, between the period 1971-2000 and 2021-2050. These results

are consistent with the Nabat et al. (2014) study, which shows that aerosol changes can explain, over the period 1980-2012, 81 ± 16% of the brightening over Europe. Concerning the impact of anthropogenic aerosols on the surface temperature increase, Figure 12 (b) shows that they can contribute of about 0.2°C on average, and up to 0.5°C locally in summer. The anthropogenic aerosol evolution explains about 6% of the expected warming over Europe on annual average with a maximum of 20% over certain regions such as the northern Black Sea in summer. The surface radiation and temperature changes between the historical

and future period over the Euro-Mediterranean region, as well as the part of these changes attributable to anthropogenic aerosols, are summarised in Table 4. In the Boé et al. (2020) and Gutiérrez et al. (2020) studies, the authors have also shown that time-varying anthropogenic aerosol forcing plays an important role on radiation and therefore temperature changes. Indeed Boé et al. (2020) showed, in average over Europe, that aerosols can explain roughly 0.5°C of the 1.8°C warming (i.e. 30% of the warming) on the 2021-2050 period with the RCP 8.5. Moreover, Figure 12 (b) also shows a strong regional variability

in terms of anthropogenic aerosols impact on surface temperature, especially during the summer season. Two maxima are simulated over Central Europe and over the Iberian Peninsula that will be studied in more detail below.

Concerning the role of anthropogenic aerosols in precipitation changes, Boé et al. (2020) show that the anthropogenic aerosol evolution could lead to an increase in precipitation over the northern half of Europe. However, in the present study, results about the impact of anthropogenic aerosols on precipitation or on surface wind speed changes showed no significant signals over the

Euro-Mediterranean region (see Figures 12 (c) and 12 (d) for details).

## 4.2  Central Europe

The summer is the season most impacted by the anthropogenic aerosol evolution as shown in the previous results. For this reason, only this season will be discussed in the rest of this study. Figures 13 and 14 show average differences of several parameters between the SSP585 and the SSP585cst simulations for the months of June, July and August over the Euro-





Mediterranean region. During the summer, the first extra-warming due to anthropogenic aerosols occurs over Central Europe and shows an extra-warming of about 0.2°C on average over this region with a maximum of 0.6°C over the north of the Black Sea (Figure 13, a). On average, this temperature increase corresponds to 10% of the expected warming over this region during this season. Figure 13 also indicates that this extra-warming over Central Europe is well correlated with a surface radiation

increase, also due to the anthropogenic aerosol evolution, of $5.8 \pm 1.2$ W m$^{-2}$ on average over this region (Figure 13, b).

Figure 13 shows that the surface solar radiation increase over Central Europe can be explained by a total aerosol direct radiative forcing decrease at the surface of the order of $4.4 \pm 0.2$ W m$^{-2}$ over this region (Figure 13, c). As shown in Figure 13 (d), this direct radiative forcing decrease over Central Europe is mainly due to the anthropogenic AOD decrease (-0.1 on average, i.e. a decrease of about 40%), largely dominated by the decrease in loads of sulfate particles. In addition, the surface

solar radiation increase over Central Europe is well correlated with the clouds optical depth (COD) decrease (of about -1.3 $\pm$ 0.3 as shown in Figure 14) (a). This change in COD is mainly due to the effective cloud droplet radius increase of about 1.2 μm over this region (Figure 14, b). The COD decrease and the effective cloud droplet radius increase over this region is caused by the sulfate concentration drop, which is illustrated by the sulfate AOD decline of about -0.14 between the historical and future periods over Central Europe. It should be reminded that in the CNRM-ALADIN63 model, only the first indirect radiative effect

of sulfate, organic carbon and sea salt aerosols is taken into account (Michou et al., 2020). The effective droplet radius and COD are therefore directly dependent on the concentration of these different aerosols types. Given the weak organic carbon and sea salt aerosol evolution between the two periods (1971-2000 and 2021-2050), changes in COD observed are therefore largely dependent on the sulfate aerosols decrease. In detail, the region of Central Europe studied here could be divided into two more specific regions where the surface solar radiation increase causes are somewhat different. Indeed, the surface solar

radiation increase over the Black Sea region is due to both the direct radiative forcing decrease and the COD decrease while over Germany, it is mostly due to the COD decrease.

Finally, the extra-warming due to anthropogenic aerosols observed over Central Europe is therefore due to a combination of aerosol-radiation (direct aerosol effect) and aerosol-cloud (indirect aerosol effect) interactions processes. The various parameters responsible for the temperature increase over this region are summarised in Figure 15.

**4.3 Iberian Peninsula and western France**

As previously mentioned, Figure 13 (a) also highlights a second extra-warming due to anthropogenic aerosols, of the order of 0.2°C, occurring over the Iberian Peninsula. This warming observed during the summer is correlated here with a surface solar radiation rise of about 4.2 W m$^{-2}$ (Figure 13, b). This increase can be explained over this region by a decrease of COD, that is of the order of -1.3 on average over the Iberian Peninsula (Figure 14, a). As mentioned previously, this change is mainly caused

by an effective droplet radius increase of about $0.8 \pm 0.1$ μm (Figure 14, b) due, as before, to the sulfate mass concentration decrease between the historical and future period. In addition, the surface solar radiation increase observed over the Iberian Peninsula can also be explained by an atmospheric dynamics change as shown in Figure 14 (c) and 16. Indeed, Figure 14 (c) shows a cloud cover decrease of -2% on average over this region which partly explains the simulated surface solar radiation increase. Figure A2, in appendix, shows that this cloud cover decrease present with the CNRM-ALADIN63 model is also found





with its forcing model (CNRM-ESM2-1) as well as with the average of the different members carried out with this ESM model. Besides, Figure 16, which shows the vertical profile of different parameters such as temperature or specific humidity obtained by the difference between SSP585 and SSP585cst simulations, also shows a surface specific humidity decrease of about 1 or 2% that is due to the atmospheric dynamics modification. This surface specific humidity decrease, which thus generates a

5 drier air in the lower layers combined with a cloud cover decrease, directly contributes to the extra-warming over the Iberian Peninsula. In summary, the extra-warming observed over the Iberian Peninsula is here due to the first indirect aerosol effect but also to an atmospheric dynamics change (semi-direct aerosol effect). Same as before, the various parameters responsible for the extra-warming over this region are summarised in Figure 17.

Finally, Figure 13 (a) shows a more particular case over western France, where a moderate surface temperature decrease

can be observed despite an extra surface radiation increase over this region (Figure 13, b). In this case, the surface temperature decrease can only be explained by a change in atmospheric dynamics (semi-direct aerosol effect), which results in a fresh air supply from the ocean to western France. The different cases studied here, during the summer for the SSP 5-8.5, clearly show that the regional contrasts in terms of anthropogenic aerosols impact on surface temperature changes between the historical and future period can be explained by various aerosol radiative effects.

## 15  5   Conclusions

This study deals with the evolution of different aerosols over the Euro-Mediterranean region between the historical period 1971-2000 and the future period 2021-2050 according to three different scenarios representing a wide range of possible futures. This study shows a total AOD decrease of about 35% over Europe which is mainly due to the sulphur dioxide emissions decrease and consequently to the sulfate AOD reduction. The study also shows that this sulfate AOD decrease is nevertheless compensated,

at 30%, by the nitrate and ammonium particles optical depth increase. These nitrate aerosols furthermore become the main contributor to total AOD and aerosol radiative forcing in the future period with a contribution of 45%. Finally, these various changes, similar for the different scenarios, lead to a total aerosol radiative forcing decrease between 1.2 and 2 W m$^{-2}$. This work thus highlights the need for climate models to take nitrate particles into account, particularly for future climate studies over the Euro-Mediterranean region.

In addition, the study of the sensitivity of the future Euro-Mediterranean climate to anthropogenic aerosols highlights the important role of such particle on shortwave radiation but also shows that they have a more moderate impact on the future temperature over this region. Indeed, this work indicates that the anthropogenic aerosol evolution could explain about 65% of the yearly shortwave radiation change between past and future periods, but this could also explain about 6%, or even 20% locally, of the expected yearly warming over Europe by mid-century for the SSP 5-8.5. This work therefore highlights that

taking into account the aerosol evolution is essential for understanding the Euro-Mediterranean future climate trends. Finally, this work also illustrates that the presence of strong regional contrasts in terms of anthropogenic aerosols impact on surface temperature can be explained by the different (direct, indirect or semi-direct) aerosol radiative forcings.



All those results show that the use of a regional climate modelling tool coupled with an interactive (natural and anthropogenic) aerosol scheme is adapted to investigate the complex anthropogenic particles feedbacks on the Euro-Mediterranean climate. Nevertheless, some assumptions such as the use of a time-constant nitric acid climatology need to be taken into account. Indeed, some studies, such as that of Cholakian et al. (2019) predict a nitrate precursors emissions decrease in the future,
which would consequently have an effect on the evolution and the climatic impact of such particles. Similarly, a better description of the indirect radiative forcing of nitrate aerosols in the CNRM-ALADIN63 model would allow a better estimation of the climatic impact of such anthropogenic particles. Besides, it is worth mentioning that the second indirect aerosol effect is not included in the present study, and is still affected by large uncertainties in regional climate modelling.

*Code and data availability.* The code of the regional climate model CNRM-ALADIN63 is available as follows: the SURFEX code is accessi-
ble using a CECILL-C Licence (http://www.cecill.info/licences/Licence_CeCILL-C_V1-en.txt) at http://www.umr-cnrm.fr/surfex; OASIS3-MCT is available at https://verc.enes.org/oasis/download; XIOS at https://forge.ipsl.jussieu.fr/ioserver and the rest of the CNRM-ALADIN63 code is available upon request to the authors. The output of the different simulations presented here are available upon request from the authors (thomas.druge@meteo.fr).

*Author contributions.* All authors designed the simulations and TD carried them out. TD prepared the paper with contributions from all
co-authors.

*Competing interests.* The authors declare that they have no conflict of interest.

*Acknowledgements.* We would like to thank Meteo-France and the Occitania region for the financial support of the first author. This work is part of the Med-CORDEX initiative (www.medcordex.eu, last access: 9 October 2020) and a contribution to the CORDEX Flagship Pilot Study (FPS) on aerosols. It also provides a contribution to the ChArMEx programme, part of the French multidisciplinary programme
MISTRALS (Mediterranean Integrated Studies aT Regional And Local Scales).



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



| | Hist | SSP119 | SSP370 | SSP585 | SSP585cst |
|---|---|---|---|---|---|
| Period of simulation | 1971-2000 | 2021-2050 | 2021-2050 | 2021-2050 | 2021-2050 |
| Scenario | / | SSP 1-1.9 | SSP 3-7.0 | SSP 5-8.5 | SSP 5-8.5 |
| Anthropogenic aerosol emissions | Historical | SSP 1-1.9 | SSP 3-7.0 | SSP5-8.5 | Historical (average) |

**Table 1.** Summary of the main characteristics of the five simulations used in this study.



| Europe | Historical | SSP 1-1.9 | SSP 3-7.0 | SSP 5-8.5 |
|---|---|---|---|---|
| Sulfate | 60 | 7 | 13 | 10 |
| Nitrate | 15 | 47.5 | 43.5 | 46 |
| Ammonium | 5 | 17 | 14.5 | 15.5 |
| Organic carbon | 6 | 6 | 8 | 8 |
| Black carbon | 2 | 1 | 1.5 | 1.5 |
| Sea salt | 7 | 12 | 10.5 | 11 |
| Dust | 5 | 9.5 | 9 | 9 |
| | | | | |
| **Mediterranean Sea** | | | | |
| Sulfate | 38.5 | 7 | 16 | 9 |
| Nitrate | 6 | 23 | 19 | 21.5 |
| Ammonium | 1.5 | 7.5 | 6 | 7 |
| Organic carbon | 4 | 3.5 | 5 | 4.5 |
| Black carbon | 1.5 | 1 | 2 | 1.5 |
| Sea salt | 14.5 | 17 | 15 | 16.5 |
| Dust | 34 | 41 | 37 | 40 |

**Table 2.** Relative contribution (%) of the different aerosols to the total AOD (550 nm) over the historical period (1971-2000) and the future period (2021-2050) according to SSPs 1-1.9, 3-7.0 and 5-8.5.



|  |  | Europe | Mediterranean Sea | Africa |
|---|---|---|---|---|
| **Shortwave** Surface | DJF | - 2.1 | - 4.3 | - 7.2 |
| | MAM | - 7.4 | - 13.1 | - 17.3 |
| | JJA | - 9.2 | - 11.9 | - 12.1 |
| | SON | - 4.2 | - 5.5 | - 5.8 |
| | **Annual mean** | **- 5.7** | **- 8.7** | **- 10.6** |
| TOA | DJF | - 1.1 | - 2.6 | 1.2 |
| | MAM | - 4.0 | - 5.9 | 5.3 |
| | JJA | - 5.3 | - 7.1 | 3.6 |
| | SON | - 2.8 | - 3.6 | 1.3 |
| | **Annual mean** | **- 3.3** | **- 4.8** | **2.8** |

**Table 3.** Mean SW DRF (W m$^{-2}$) of all aerosols over the period 1971-2000 at the surface and at the TOA over Europe, Mediterranean Sea and Africa.





|  |  | Europe | | Mediterranean Sea | | Africa | |
|---|---|---|---|---|---|---|---|
|  |  | SSP585 - Hist | SSP585 - SSP585cst | SSP585 - Hist | SSP585 - SSP585cst | SSP585 - Hist | SSP585 - SSP585cst |
| Radiation | DJF | 5.3 | 2.4 (45%) | 3.0 | 1.6 (53%) | - 1.3 | - 0.2 (15%) |
|  | MAM | 5.8 | 4.1 (71%) | 1.6 | 1.4 (87%) | - 2.2 | - 0.6 (27%) |
|  | JJA | 7.5 | 4.7 (63%) | 2.5 | 3.5 (140%) | - 2.6 | - 0.3 (11%) |
|  | SON | 3.2 | 3.0 (94%) | 1.6 | 1.5 (94%) | - 1.8 | - 0.4 (20%) |
|  | **Annual mean** | **5.4** | **3.5 (65%)** | **2.2** | **2.0 (91%)** | **- 2.0** | **- 0.4 (20%)** |
| Temperature | DJF | 2.5 | 0.12 (5%) | 1.7 | - 0.005 (- 0.3%) | 1.6 | 0.01 (0.6%) |
|  | MAM | 1.7 | 0.12 (7%) | 1.5 | 0.002 (0.1%) | 1.8 | 0.02 (1.1%) |
|  | JJA | 2.3 | 0.16 (7%) | 1.8 | 0.01 (0.6%) | 2.0 | 0.06 (3%) |
|  | SON | 2.1 | 0.12 (6%) | 1.7 | - 0.006 (- 0.3%) | 1.9 | 0.02 (1%) |
|  | **Annual mean** | **2.2** | **0.13 (6%)** | **1.7** | **0.001 (0.1%)** | **1.9** | **0.03 (1.5%)** |

**Table 4.** Average differences between the SSP585 and Hist simulations showing the surface radiation (W m$^{-2}$) and the surface temperature (°C) increase over Europe, the Mediterranean Sea and Africa between the period 1971-2000 and 2021-2050 and between the SSP585 and SSP585cst simulations showing the aerosols contribution to this brightening (W m$^{-2}$ and %) and to this surface warming (°C and %).



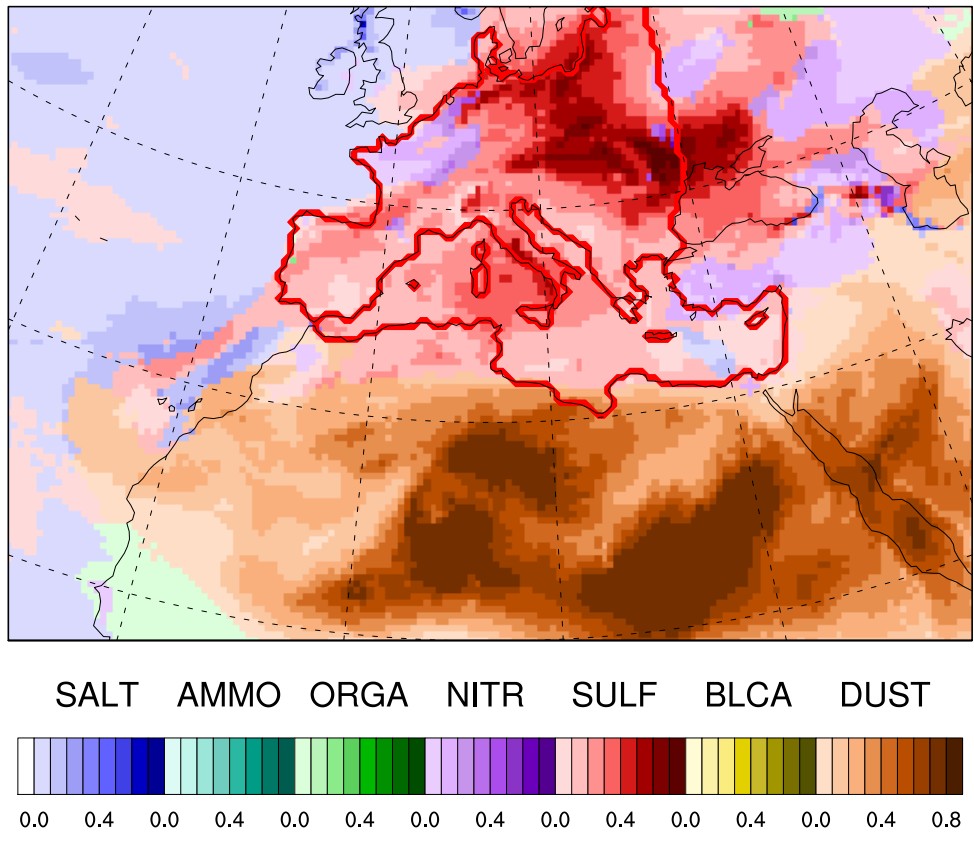

**Figure 1.** AOD of the dominant aerosol species (Sea salt, ammonium, organic carbon, nitrate, sulfate, black carbon and dust) at each point over the Euro-Mediterranean region (28th August 1974). The inner model domain, without the bi-periodization and the relaxation zones, represents 101 x 153 points. The projection type used here is the Lambert conformal projection. The two main zones used in this study, the Europe and the Mediterranean Sea, are highlighted in red.





**Figure 2.** CMIP5 and CMIP6 emissions $(\mathrm{kg\,m^{-2}\,s^{-1}})$, over Europe, from the main activity sectors (residential, agriculture, energy, industrial, air, land and sea transport, waste and solvents) for sulphur dioxide, ammonia, black carbon and organic carbon. The large filled dots symbolize the 3 scenarios chosen for this study (SSP 1-1.9, SSP 3-7.0 and SSP 5-8.5). Shaded bars highlight the past and future simulations period.

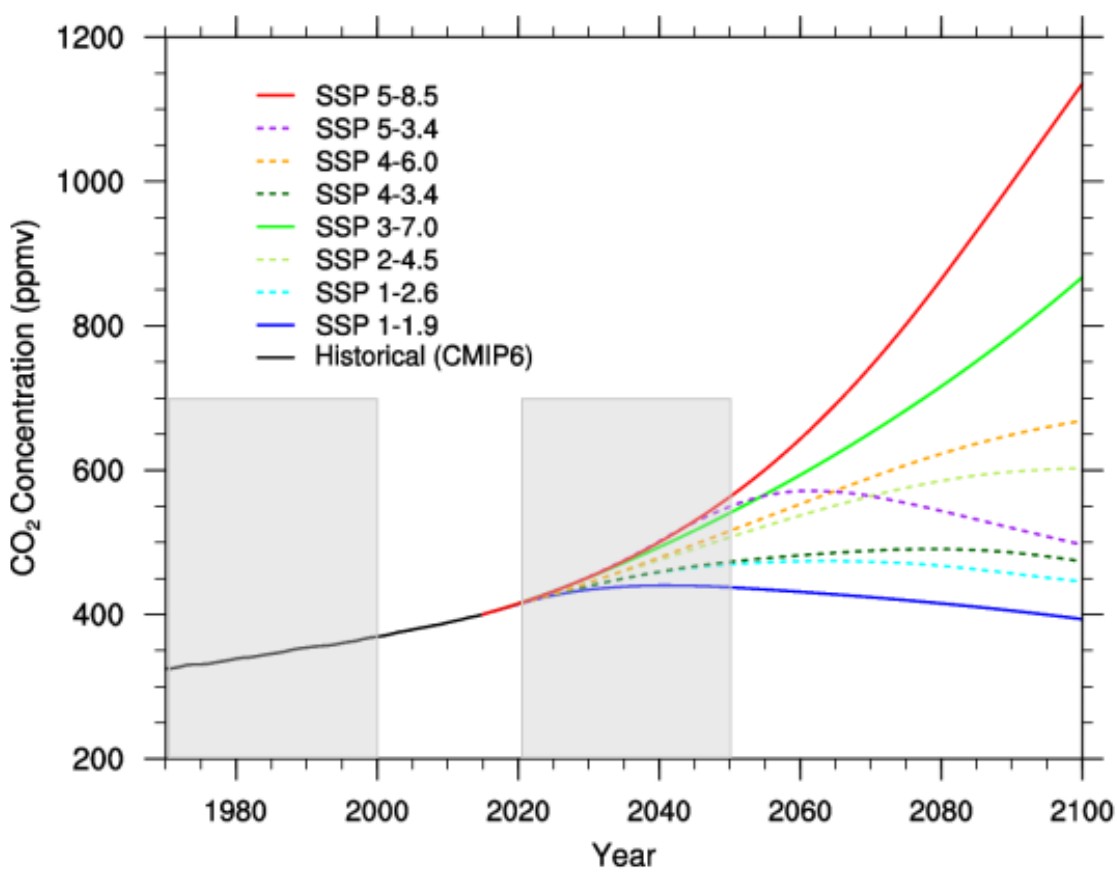

**Figure 3.** Global $CO_2$ concentration evolution (ppmv), over the period 1970-2100, according to 8 different SSPs. The 3 scenarios chosen for this study (SSP 1-1.9, SSP 3-7.0 and SSP 5-8.5) are highlighted by full lines. Shaded bars highlight the past and future simulations period.





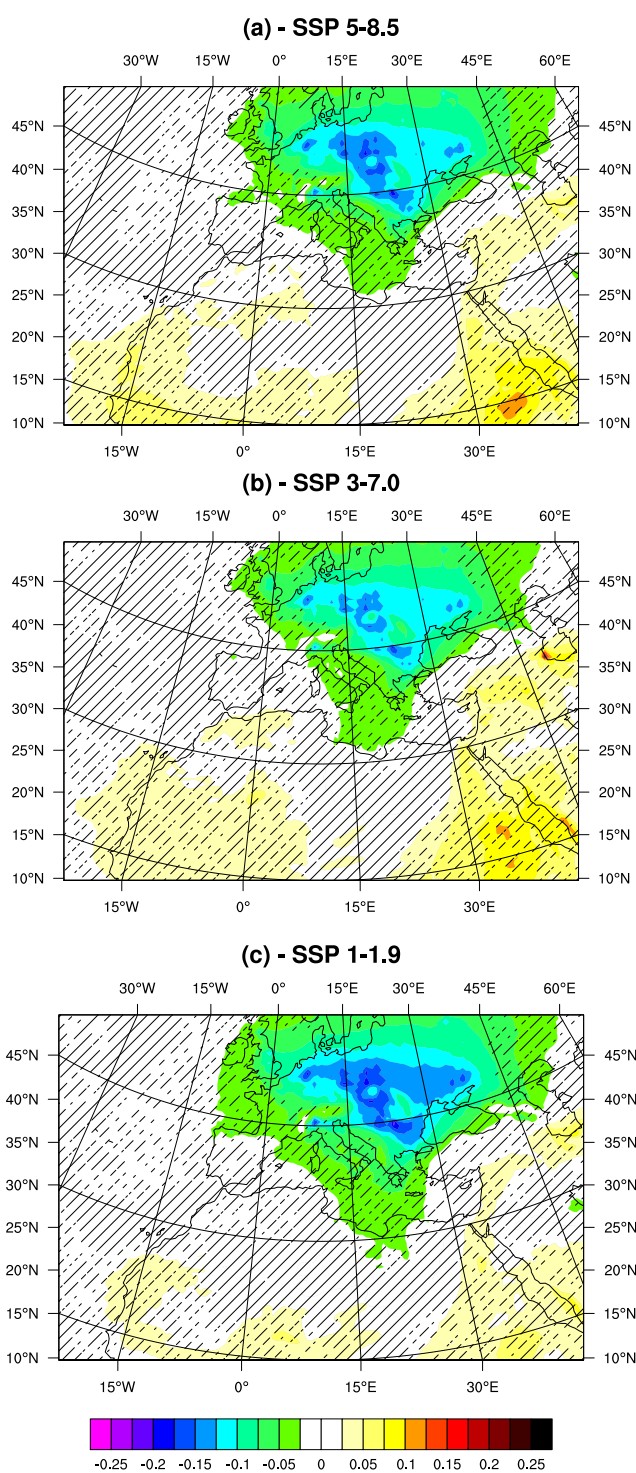

**Figure 4.** Total AOD evolution between the past period (1971-2000) and the future period (2021-2050) following SSP 5-8.5 (a), SSP 3-7.0 (b) and SSP 1-1.9 (c). The hatched areas are statically non significant with a threshold of 10%.



**Figure 5.** Surface concentration ($\mu g\,m^{-3}$) and AOD evolution of the different aerosols between the past period (1971-2000) and the future period (2021-2050) according to SSP 1-1.9, SSP 3-7.0 and SSP 5-8.5 over Europe (a) and the Mediterranean Sea (b).





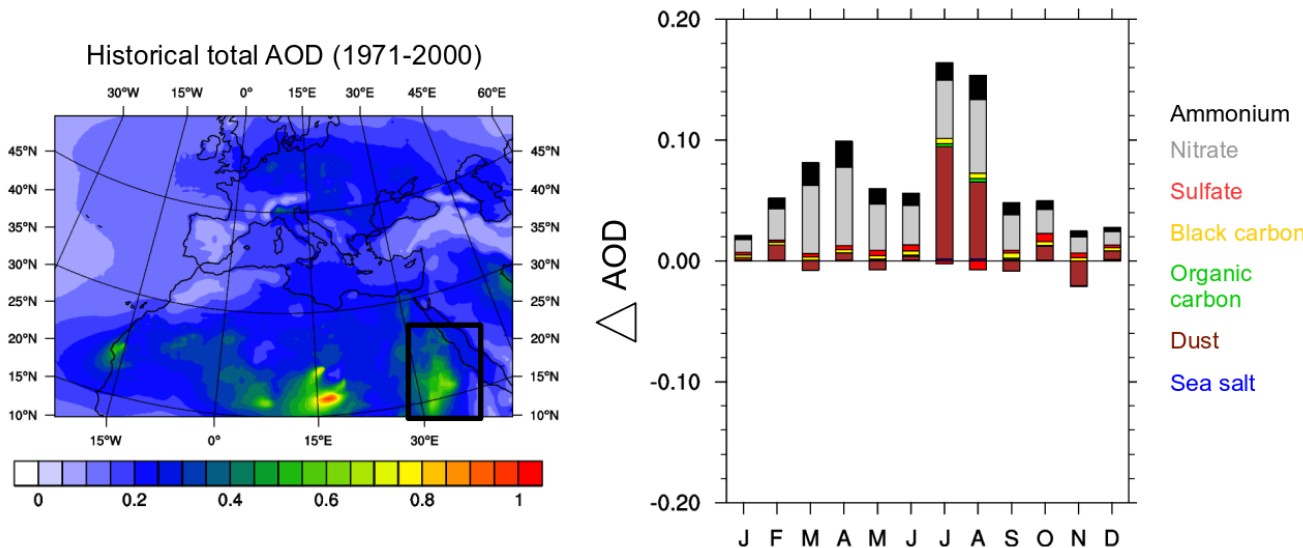

**Figure 6.** Historical total AOD and its evolution between the past period (1971-2000) and the future period (2021-2050) according to the SSP 5-8.5 over the East African region which is framed in black. The hatched areas are statically non significant with a threshold of 10%.





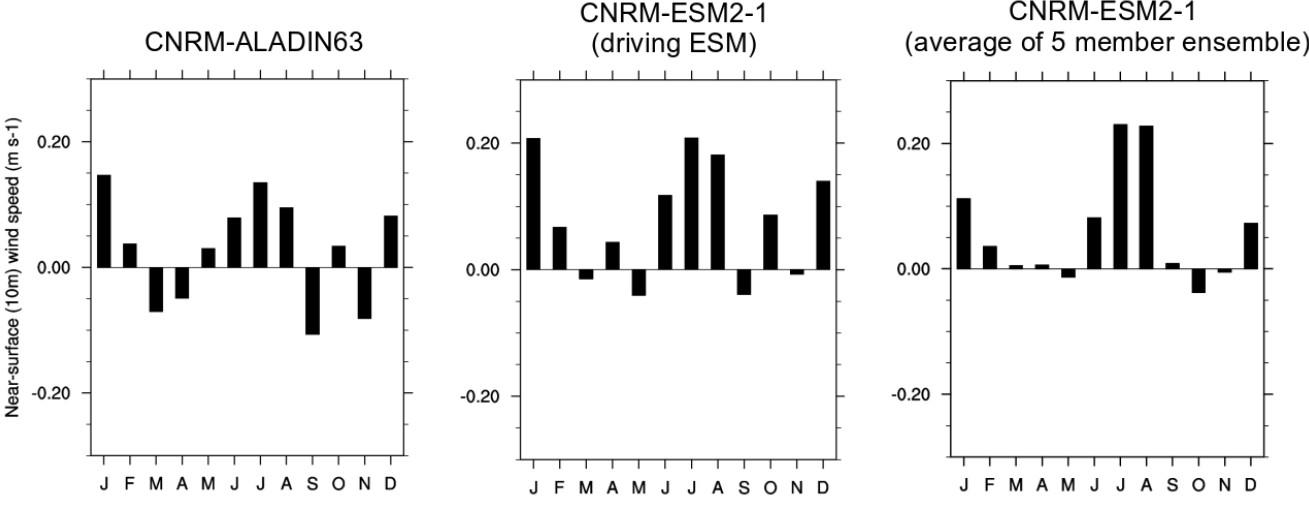

**Figure 7.** Near surface wind speed evolution (m s$^{-1}$) between the past period (1971-2000) and the future period (2021-2050) according to the SSP 5-8.5 over the East African region with the CNRM-ALADIN63 model and its driving ESM.



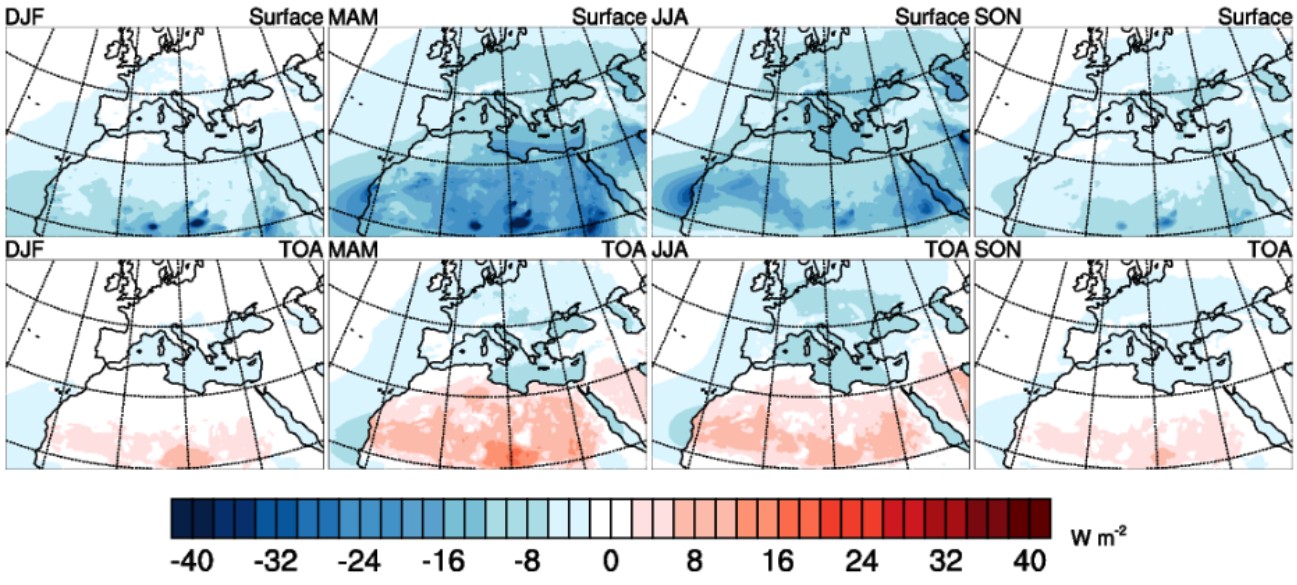

**Figure 8.** Mean SW DRF (W m$^{-2}$) of all aerosols over the period 1971-2000 at the surface and at the TOA.





**Figure 9.** Mean aerosols SW DRF evolution (W m$^{-2}$) at the surface and at the TOA between the historical period (1971-2000) and the future period (2021-2050) according to SSPs 5-8.5 (a), 3-7.0 (b) and 1-1.9 (c). The hatched areas are statically non significant with a threshold of 10%.



**Figure 10.** Surface SW DRF evolution (W m$^{-2}$) of sulfate and organic carbon (in red), sea salt (in blue), dust (in brown), black carbon (in yellow) and nitrate and ammonium (in grey) between the historical period (1971-2000) and the future period (2021-2050) according to SSPs 1-1.9, 3-7.0 and 5-8.5.





**Figure 11.** Surface radiation (W m$^{-2}$, a), surface temperature (°C, b), precipitation (mm day$^{-1}$, c) and surface wind speed (m s$^{-1}$, d) evolution between the period 1971-2000 and 2021-2050 for the SSP 5-8.5. The hatched areas are statically non significant with a threshold of 10%.





**Figure 12.** Anthropogenic aerosols role (difference between SSP585 and SSP585cst) in surface radiation (a), surface temperature (b), precipitation (c) and surface wind speed (d) changes between the period 1971-2000 and 2021-2050. The hatched areas are statically non significant with a threshold of 10%.

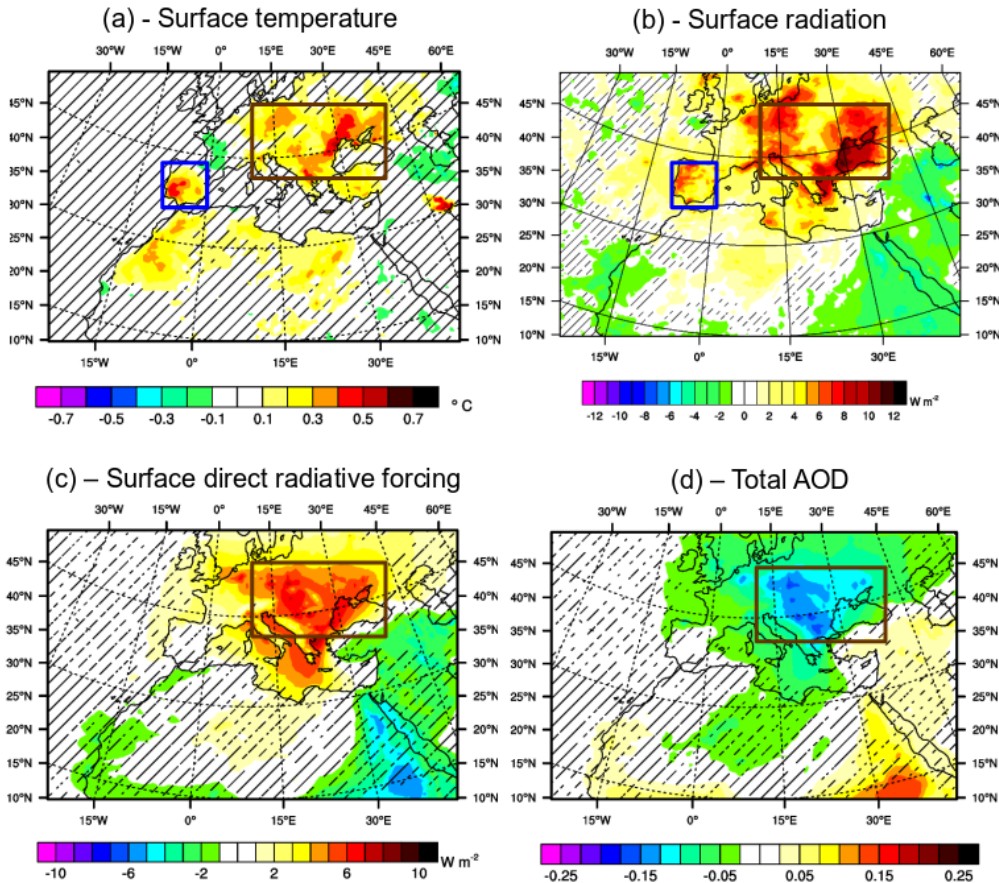

**Figure 13.** Mean differences, for the months of June, July and August, between SSP585 and SSP585cst simulations over the period 2021-2050 for the surface temperature (°C, a), the surface radiation (W m$^{-2}$, b), the surface direct radiative forcing (W m$^{-2}$, c) and the total AOD (d). The brown frame represents Central Europe and the blue frame locates the Iberian Peninsula. The hatched areas are statically non significant with a threshold of 10%.

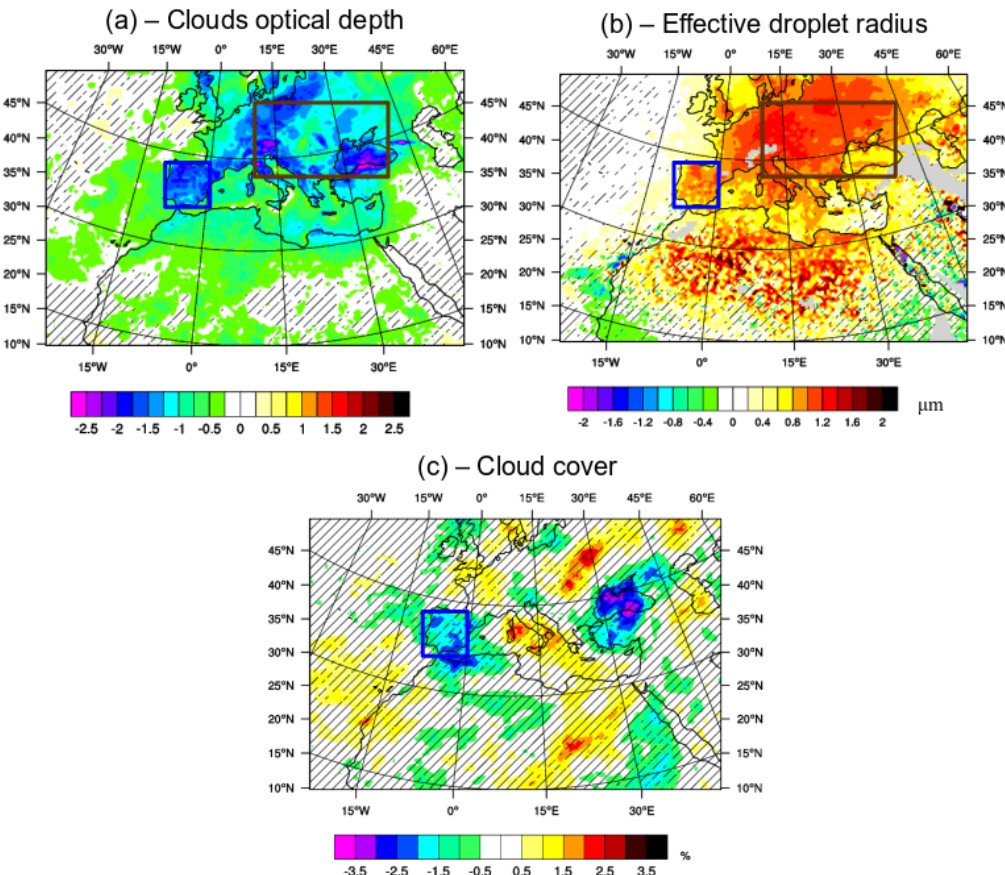

**Figure 14.** Mean differences, for the months of June, July and August, between SSP585 and SSP585cst simulations over the period 2021-2050 for the cloud optical depth (a), the effective droplets radius (μm, b) and the cloud cover (%, c). The brown frame represents Central Europe and the blue frame locates the Iberian Peninsula. The hatched areas are statically non significant with a threshold of 10%.



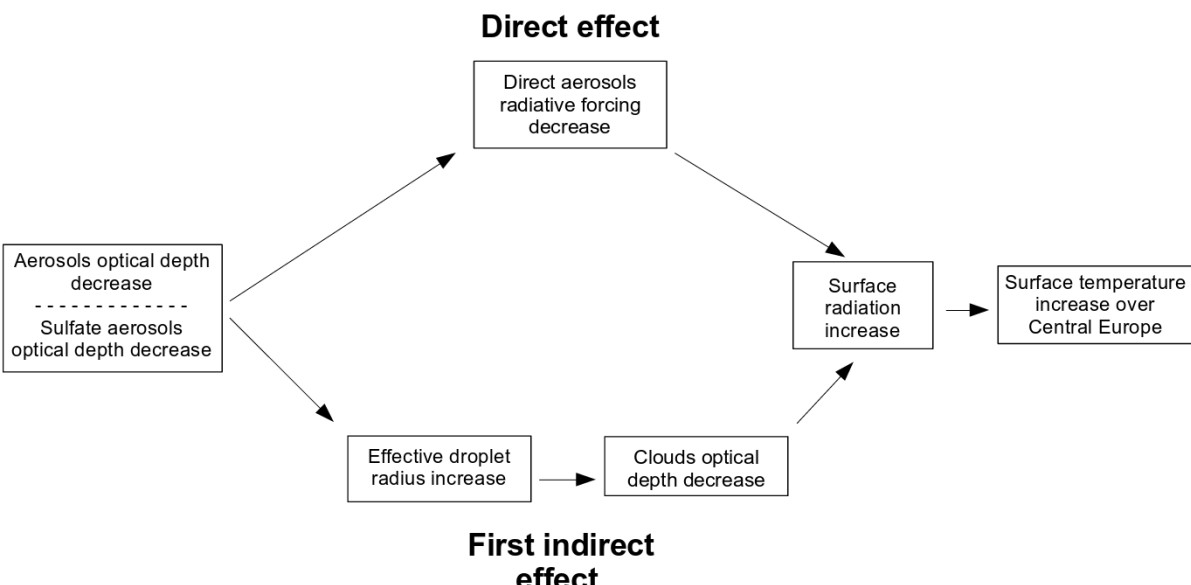

**Figure 15.** Summary of the anthropogenic aerosols role on the projected surface temperature increase over Central Europe in summer between the period 1971-2000 and 2021-2050.



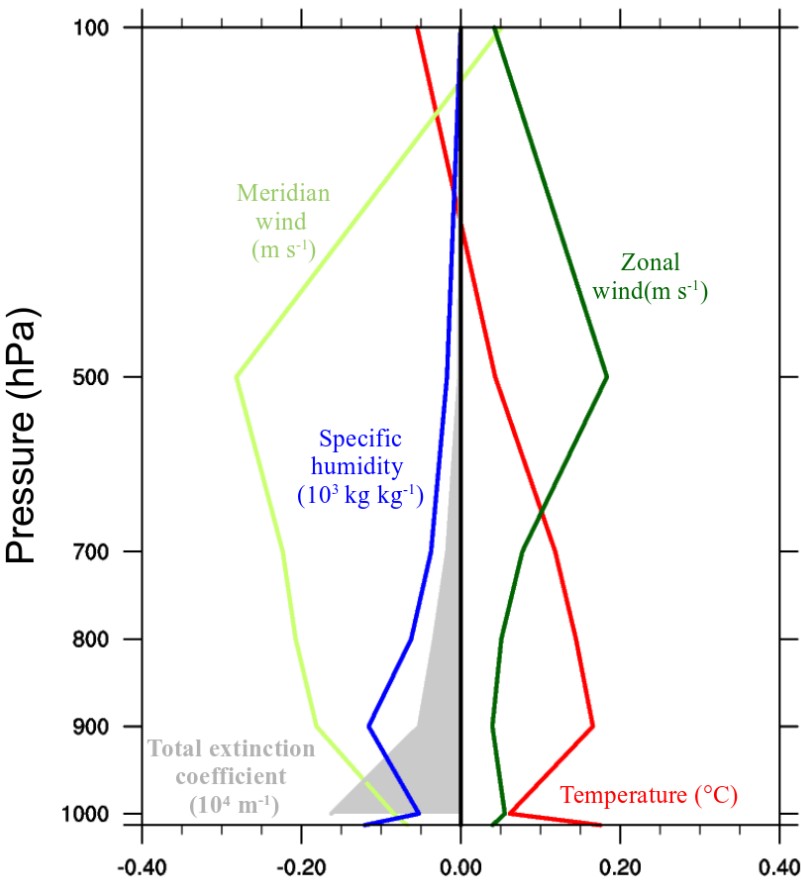

**Figure 16.** Vertical profile, obtained by the difference between SSP585 and SSP585cst simulations, of the temperature (°C), the total extinction coefficient ($10^4$ m$^{-1}$), the meridional and zonal wind (m s$^{-1}$) and the specific humidity ($10^3$ Kg Kg$^{-1}$) over the Iberian Peninsula for the months of June, July and August.





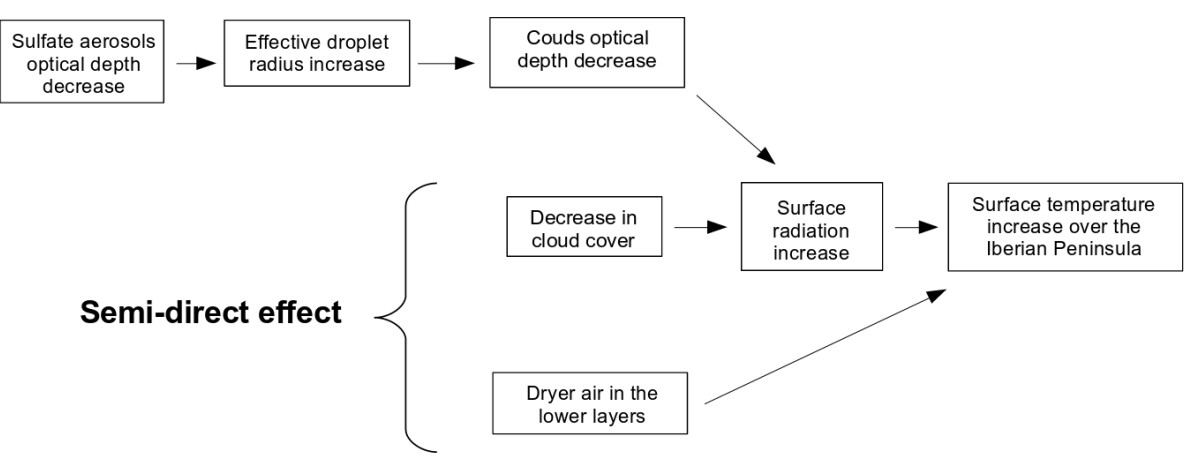

**Figure 17.** Summary of the anthropogenic aerosols role on the projected surface temperature increase over the Iberian Peninsula in summer between the period 1971-2000 and 2021-2050.





**Appendix A:**

**Figure A1.** SW DRF evolution (W m$^{-2}$), at the TOA, of sulfate and organic carbon (in red), sea salt (in blue), dust (in brown), black carbon (in yellow) and nitrate and ammonium (in grey) between the historical period (1971-2000) and the future period (2021-2050) according to SSPs 1-1.9, 3-7.0 and 5-8.5.



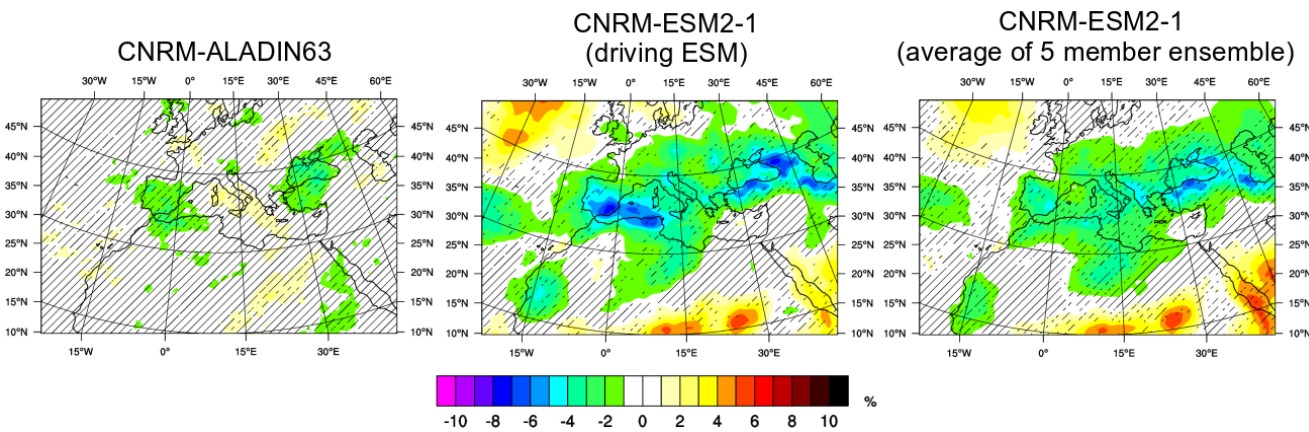

**Figure A2.** Mean difference, for the months of June, July and August, between SSP585 and SSP585cst simulations over the period 2021-2050 for the cloud cover (%) with the CNRM-ALADIN63 model and its driving ESM. The hatched areas are statically non significant with a threshold of 10%.