# Peer review of "Future evolution of aerosols and implications for climate change in the Euro-Mediterranean region using the CNRM-ALADIN63 regional climate model"

_Atmospheric Chemistry and Physics, 2020_

## Referee Comment (RC1) · Anonymous Referee #1 · 4 Jan 2021

This manuscript investigates the near future evolution of aerosols and their implications for climate change in the Euro-Mediterranean using a regional climate model. It is a very interesting study showing original results while it is generally well written and presented. I suggest acceptance of the manuscript for publication but I have a number of comments that have to be addressed before the final acceptance.

Comments

1) Page 1, lines 13-15: Maybe it would be to mention in the abstract what is the range of sulfate and nitrate and ammonium DRF for the different scenarios. 2) Page 2, lines 21-25: Besides the advantages of resolution to study aerosol-climate interactions with RCMs, it is also important to mention the limitations in comparison to Earth System Models (ESMs). For example, can the regional model domain setup account a) for the slow climate responses to aerosols and b) the influence of remote forcings of aerosols simulated in global models? Are the climate responses due to aerosols in RCMs more comparable with the fast responses simulated in ESMs when SSTs are prescribed? 3) Page 3, lines 5-7: A reference could be added here. 4) Page 3, lines 33-35: This is an important point as there is limited number of RCM studies that account for aerosol-cloud interactions. See for example a recent study by Pavlidis et al. ( https://doi.org/10.5194/gmd-13-2511-2020). 5) Page 5, lines 20-22: I am rather confused. What about the ammonium and nitrate aerosol precursors in the future simulations? Is it based again on CAMS climatology? Please clarify accordingly within the manuscript. 6) Page 6, lines 9-10: This is further supported from the fact that non-methane mitigation of short-lived climate forcers leads to a net warming effect in the near-term due to the removal of aerosol (see e.g. Allen et al., 2020, https://doi.org/10.5194/acp-20-9641-2020). 7) Page 8, lines7: Figure 4 shows the CNRM-ALADIN63 AOD results. How that compares with CNRM-ESM2-1 as both share the same physics and same forcings and include the TACTIC aerosol scheme (except nitrate and ammonium particles that are only used in CNRM-ALADIN63). The authors make already a comparison between the RCM and the driving ESM for cloud cover in Figure A2 but maybe a comparison for AOD and SW direct RF would be also useful. 8) Page 8, lines 9-12: It would be useful to add a sentence on how the modelled aerosol contributions to AOD over the historical period in Figure 5 compare with observational studies based on satellite data. For example despite the different periods, I see a reasonable consistency with contribution of different aerosol types to the aerosol optical depth based on MODIS over Mediterranean in Figure 7 by Gergoulias et al.( doi:10.5194/acp-16-13853-2016). 9) Page 8, lines 23-25: Over Mediterranean, at least in surface concentration for a few months, it seems there are some important contributions from dust and sea salt changes under SSPs (Figure 5). Also in AOD there are small but notable changes due to dust changes under SSPs in a few months.

10) Page 10, lines 5-6: So maybe you can specify that this is an instantaneous radiative forcing. 11) Page 10, line 1: The authors give emphasis on the direct radiative forcing but skip the cloud radiative forcing and the forcing due to semi-direct effects. Could they justify why not discussing them? 12) Page 12, lines 3-5: Please be more specific for which months and scenario. 13) Page 14, lines 10: Please can you discuss quantitatively what is well correlated? 14) Page 14, lines 19-21: The authors discuss the impact of cloud optical depth on surface sola radiation. However, I was wondering why not calculating and discussing a cloud RF. 15) Page 14, lines 32: Maybe in order to strengthen the possibility that atmospheric dynamics contributes to surface solar radiation increase it would be nice to look also circulation changes (e.g. mslp changes). Maybe there is an anticyclonic circulation anomaly, Cloud cover itself in Figure 14c is not so straightforward indicator. Also the cloud cover decrease over Iberian Peninsula is related to the decrease of COD. 16) Page 15, lines 7: Maybe you have to check if there is present an anticyclonic circulation anomaly that plays a role on what is shown in Figure 16.

---

## Referee Comment (RC2) · Anonymous Referee #2 · 4 Feb 2021

The paper analyses the differences in aerosol load and DRF between the end of the 20th century (1971-2000) and the mid-21st century (2020-2050) in the Euro-Mediterranean region using a regional climate model (CNRM-ALADIN63) coupled to the TACTIC (Tropospheric Aerosols for ClimaTe In CNRM) interactive aerosol scheme and driven by the global CNRM-ESM2-1 Earth System Model (used in CMIP6).

The study reports the already well known decrease in sulfate and increase in nitrate between the two periods, estimating a DRF decrease of 2.6 W/m2 and increase of 1.4 W/m2 respectively. The study also concludes that the extra-warming attributable to the anthropogenic aerosols evolution over Central Europe and the Iberian Peninsula (0.2°

[Figure]

C) during the summer period is due to "aerosol-radiation" as well as "aerosol-cloud" interactions processes.

The study is interesting and deserves publication but I have some general comments that should be convincingly addressed before a final decision on the manuscript is made:

- The title: I believe the title is slightly misleading. The study adresses the implications of the aerosol evolution upon climate change in the Euro-Mediterranean region using one regional model driven by one ESM. I also note that, despite the higher resolution of the model compared to most ESM's, it includes several simplifications (omission of processes and simplified schemes) that do not necessarily represent the state-of-the-art when it comes to the interaction of aerosols with radiation and clouds. I believe the title should clearly reflect that the study focuses in one model (therefore implicitly conveying the more correct message that the results may be to some extent model- and assumption-dependent). As it stands the reader may expect a multi-model study including uncertainties, which is not the case. For example: "Future evolution of aerosols and implications for climate change in the Euro-Mediterranean region using the CNRM-ALADIN63 regional climate model" or something similar would be more appropriate.

- I think it is important that the authors nuance a bit more their statements in the introduction about the study of aerosol effects with a regional climate model. While resolution is important, the resolution used is not far from what some ESM models are already using. More importantly, there are other aspects that are key to understand aerosol impacts and at present many ESMs already include aerosol and cloud processes that are far more advanced than the ones represented in this study. The regional model used here includes a simplified aerosol scheme (without aerosol microphysics and number, for example), and many other simplifications (like introducing only the first indirect effect or using a constant nitric acid climatology). Besides that, regional climate modeling cannot account for slow climate responses to aerosols in the domain, and in fact heavily depends upon the ESM driver through the boundaries.

[Figure]

I think these aspects and the associated limitations should be clearly explained and discussed both in the introduction and the discussion.

- Nitric acid is implemented in the model as a constant monthly climatology based on the CAMS reanalysis. One key result reflected in the abstract is the increase in nitrate and its impact upon the DRF. At least, the abstract should clearly acknowledge the constant nitric acid assumption.

- In page 6 you state: "The future period has been selected in the near future because, unlike greenhouse gases, the most important aerosol change is up to the middle of the century. Moreover, the near future horizon period is most suitable to help public decision-makers." I agree with this statement. However, I find that the selection of the reference period (1971-200) is not well justified and may even been a bit inconsistent with the argument that "is most suitable to help public decision makers". Why the reference period is 1971-2000. Is it because of sulfate and the associated large signal in the DRF? Wouldn't policy makers prefer to see the differences between mid-century and the present day?

- Given that you are using a regional climate model driven by an ESM with a similar (although not identical) aerosol scheme, they should be compared. In fact, average results of the DRF are compared to other studies throughout the paper but I really miss a clear and consistent comparison with the parent ESM. This could also respond to the question: how useful is resolution and downscaling for the diagnostics that are discussed in the paper? It would be even better if other available CMIP6 ESM results could be consistently compared within exactly the same domain. That would provide a solid comparison reference.

- Please clearly state that you are calculating the instantaneous RF. Current practice is to calculate effective radiative forcing which includes the forcing and the fast response. Also, why only the direct radiative forcing is discussed in section 3.2 and not the indirect effect? What is the relative role of both upon the total forcing? This is particularly

strange as in section 4 you discuss on the interaction with clouds and you suggest influences of the indirect effect upon the certain results.

---

## Author Comment (AC1) · 15 Mar 2021

We would like to thank the two anonymous referees for their comments mentioning different points listed below.

**Anonymous Referee 1**

This manuscript investigates the near future evolution of aerosols and their implications for climate change in the Euro-Mediterranean using a regional climate model. It is

a very interesting study showing original results while it is generally well written and presented. I suggest acceptance of the manuscript for publication but I have a number of comments that have to be addressed before the final acceptance.

1- Page 1, lines 13-15: Maybe it would be to mention in the abstract what is the range of sulfate and nitrate and ammonium DRF for the different scenarios.
The range of sulfate and nitrate and ammonium DRF for the different scenarios has been added: "The different evolution of aerosols therefore impacts their DRF, with a significant sulfate DRF decrease between 2.4 and 2.8 W m$^{-2}$ and a moderate nitrate and ammonium DRF increase between 1.3 and 1.5 W m$^{-2}$, depending on the three scenarios over Europe." (page 1, lines 15-16).

2- Page 2, lines 21-25: Besides the advantages of resolution to study aerosol-climate interactions with RCMs, it is also important to mention the limitations in comparison to Earth System Models (ESMs). For example, can the regional model domain setup account a) for the slow climate responses to aerosols and b) the influence of remote forcings of aerosols simulated in global models? Are the climate responses due to aerosols in RCMs more comparable with the fast responses simulated in ESMs when SSTs are prescribed?
As the chosen domain is assumed to be large enough to include the main sources of aerosols over the Euro-Mediterranean region, no aerosols are transmitted from the global to the regional model CNRM-ALADIN63 (page 7, lines 4-6). The study of long-range aerosol transport (fires) is not the objective of this study. We focus here on local aerosols. Moreover, in this study, the aerosols effects on SST are not taken into account, we therefore focus mainly on rapid climate responses to aerosol forcings (Page 4, lines 18-19).

3- Page 3, lines 5-7: A reference could be added here.

The reference "Nabat et al. 2015" has been added (page 3, line 19).

4- Page 3, lines 33-35: This is an important point as there is limited number of RCM studies that account for aerosol-cloud interactions. See for example a recent study by Pavlidis et al. (https://doi.org/10.5194/gmd-13-2511-2020).
We acknowledge aerosol-cloud interactions are an important point in climate-aerosol studies. The reference as well as the main results of the study have been added in the introduction (page 2 line 31 and page 3 lines 11-17).

5- Page 5, lines 20-22: I am rather confused. What about the ammonium and nitrate aerosol precursors in the future simulations? Is it based again on CAMS climatology? Please clarify accordingly within the manuscript.
In future simulations, ammonia emissions come from CMIP6 dataset (Gidden et al. 2019), as presented in Figure 2 in the manuscript. Concerning the nitric acid, we always use a constant monthly $HNO_3$ climatology calculated from CAMS climatology (page 5 lines 32-33). This is indeed an important limitation in this study. Figure Appendix B1 (will not be shown in the article) presents the $HNO_3$ evolution in 4 CMIP6 models and in the CAMS reanalysis. There is indeed a $HNO_3$ decrease in the future period in the lower atmospheric layers (925 and 850 hPa). On the other hand, at 700 hPa this decrease is less visible. Moreover, given the differences between the models, the uncertainty remains relatively high. When nitrate aerosols have been included in the TACTIC aerosol scheme, we performed a sensitivity test to evaluate the impacts of a time-dependent or a constant nitric acid climatology. Figure Appendix B2 (will not be shown in the article) shows that, in our model, the nitrate concentration is relatively little impacted by the use of a constant or time-dependent nitric acid climatology. For this reason, we decided to use a constant nitric acid climatology in this version of the model (page 5, lines 30-33). Nevertheless, the implementation of a time-dependent nitric acid climatology is envisaged in a future version of the TACTIC aerosol scheme

(page 17 lines 5-6).

6- Page 6, lines 9-10: This is further supported from the fact that non-methane mitigation of short-lived climate forcers leads to a net warming effect in the near-term due to the removal of aerosol (see e.g. Allen et al.,2020, https://doi.org/10.5194/acp-20-9641-2020).
The reference and the sentence: "In their study, Allen et al. (2020) have effectively shown that non-methane mitigation leads to a net warming effect in the near-term due to the removal of aerosol" have been added (page 6 lines 19-20).

7- Page 8, lines 7: Figure 4 shows the CNRM-ALADIN63 AOD results. How that compares with CNRM-ESM2-1 as both share the same physics and same forcings and include the TACTIC aerosol scheme (except nitrate and ammonium particles that are only used in CNRM-ALADIN63). The authors make already a comparison between the RCM and the driving ESM for cloud cover in Figure A2 but maybe a comparison for AOD and SW direct RF would be also useful.
A figure showing the comparison between the RCM and the driving ESM for AOD and surface SW DRF has been added in Appendix (Figure A1 and Figure A2). The forcing model shows a similar AOD and surface SW DRF trends over Europe and the Mediterranean Sea (page 8, lines 27-29 and page 11, lines 31-34). However, we can observe a surface SW DRF decrease more pronounced with the CNRM-ESM2-1 global model due to the fact that nitrate and ammonium aerosols are not taken into account in this model (page 11, lines 31-34).

8- Page 8, lines 9-12: It would be useful to add a sentence on how the modelled aerosol contributions to AOD over the historical period in Figure 5 compare with observational studies based on satellite data. For example despite the different periods, I see a reasonable consistency with contribution of different aerosol types

to the aerosol optical depth based on MODIS over Mediterranean in Figure 7 by Gergoulias et al. (doi:10.5194/acp-16-13853-2016).

The reference and a sentence describing the results of the study have been added: "Concerning the Mediterranean Sea, Table 2 indicates a historical natural (sea salt and dust) aerosols contribution to the total AOD of about 50%. These results are consistent with the study of Georgoulias et al. (2016) carried out over the Eastern Mediterranean and based on MODIS Aqua (2002-2012) and MODIS Terra (2000-2012) observations, that shows a dust and sea salt aerosols contribution to the total AOD (550 nm) of about 60%" (page 10 lines 6-9).

9- Page 8, lines 23-25: Over Mediterranean, at least in surface concentration for a few months, it seems there are some important contributions from dust and sea salt changes under SSPs (Figure 5). Also in AOD there are small but notable changes due to dust changes under SSPs in a few months.

A sentence has been added to the text to mention the dust and sea salt concentration changes over the Mediterranean Sea (page 9 lines 6-7).

10- Page 10, lines 5-6: So maybe you can specify that this is an instantaneous radiative forcing.

Done (page 10, line 26).

11- Page 10, line 1: The authors give emphasis on the direct radiative forcing but skip the cloud radiative forcing and the forcing due to semi-direct effects. Could they justify why not discussing them?

Studying the cloud radiative forcing and the forcing due to semi-direct effects is effectively not the main objective of this work. Moreover, the indirect aerosol radiative effect is not fully taken into account (only the first indirect effect and nitrate aerosols are not yet included). On this specific aspect, we are waiting for more literature to be

available before including nitrate aerosols in the first indirect effect. On the other hand, the cloud radiative forcing and the forcing due to semi-direct effects have nevertheless been studied for some specific regions (as central Europe, Iberian peninsula) in a qualitative way with the different mechanisms involved (sections 4.2 and 4.3).

12- Page 12, lines 3-5: Please be more specific for which months and scenario.
These details have been added (page 12, lines 29-32).

13- Page 14, lines 10: Please can you discuss quantitatively what is well correlated?
The percentage of the surface solar radiation increase and that of the clouds optical depth decrease have been added (page 15, lines 6-7).

14- Page 14, lines 19-21: The authors discuss the impact of cloud optical depth on surface solar radiation. However, I was wondering why not calculating and discussing a cloud RF.
To complete this study, we have now added a figure presenting the effective radiative forcing (ERF) at the Top Of Atmosphere (TOA), due to aerosol-radiation interactions (ERFari, Figure 13 (d)) and to aerosol-cloud interactions (ERFaci, Figure 14 (b)). These figures are discussed in the text (page 1, lines 22 and 24, page 15, lines 1-4 and page 15 lines 26-27).

15- Page 14, lines 32: Maybe in order to strengthen the possibility that atmospheric dynamics contributes to surface solar radiation increase it would be nice to look also circulation changes (e.g. mslp changes). Maybe there is an anticyclonic circulation anomaly, Cloud cover itself in Figure 14c is not so straightforward indicator. Also the cloud cover decrease over Iberian Peninsula is related to the decrease of COD.
We have added a figure (Figure14 (e)) presenting the sea-level pressure difference

between SSP585 and SSP585cst simulations. This figure effectively shows the presence of an anticyclonic circulation anomaly over the near Atlantic which may explain in part the cloud cover and the surface specific humidity decrease over the Iberian Peninsula (page 16, lines 4-6).

16- Page 15, lines 7: Maybe you have to check if there is present an anticyclonic circulation anomaly that plays a role on what is shown in Figure 16.
See answer of the comment 15.

[Figure]

**Fig. 1.** B1 HNO3 evolution in 4 CMIP6 models and in the CAMS reanalysis over the period 1971-2050 at 925, 850 and 700 hPa (not shown in the article)

[Figure]

**Fig. 2.** B2 Nitrate concentration evolution (fine bin on the left and coarse bin in the right) over Europe using a time-dependent (red) or constant (black) hno3 climatology. The blue dotted line was used for a

[Figure]

**Fig. 3.** A1 Total AOD evolution between the past period (1971-2000) and the future period (2021-2050) with the CNRM-ALADIN63 model and its driving ESM.

[Figure]

**Fig. 4.** A2 Surface SW DRF evolution between the past period (1971-2000) and the future period (2021-2050) with the CNRM-ALADIN63 model and its driving ESM.

[Figure]

**Fig. 5.** Figure 13 Mean differences, for the months of June, July and August, between SSP585 and SSP585cst simulations over the period 2021-2050 for different parameters

[Figure]

**Fig. 6.** Figure 14 Mean differences, for the months of June, July and August, between SSP585 and SSP585cst simulations over the period 2021-2050 for different parameters

---

## Author Comment (AC2) · 15 Mar 2021

We would like to thank the two anonymous referees for their comments mentioning different points listed below.

**Anonymous Referee 2**

The paper analyses the differences in aerosol load and DRF between the end of the 20th century (1971-2000) and the mid-21st century (2020-2050) in the Euro-

Mediterranean region using a regional climate model (CNRM-ALADIN63) coupled to the TACTIC (Tropospheric Aerosols for ClimaTe In CNRM) interactive aerosol scheme and driven by the global CNRM-ESM2-1 Earth System Model (used in CMIP6). The study reports the already well known decrease in sulfate and increase in nitrate between the two periods, estimating a DRF decrease of 2.6 W/m2 and increase of 1.4W/m2 respectively. The study also concludes that the extra-warming attributable to the anthropogenic aerosols evolution over Central Europe and the Iberian Peninsula (0.2âŬęC) during the summer period is due to "aerosol-radiation" as well as "aerosol-cloud"interactions processes. The study is interesting and deserves publication but I have some general comments that should be convincingly addressed before a final decision on the manuscript is made:

1- The title: I believe the title is slightly misleading. The study adresses the implications of the aerosol evolution upon climate change in the Euro-Mediterranean region using one regional model driven by one ESM. I also note that, despite the higher resolution of the model compared to most ESM‒s, it includes several simplifications (omission of processes and simplified schemes) that do not necessarily represent the state-of-the-art when it comes to the interaction of aerosols with radiation and clouds. I believe the title should clearly reflect that the study focuses in one model (therefore implicitly conveying the more correct message that the results may be to some extent model- and assumption-dependent). As it stands the reader may expect a multi-model study including uncertainties, which is not the case. For example: "Future evolution of aerosols and implications for climate change in the Euro-Mediterranean region using the CNRM-ALADIN63 regional climate model" or something similar would be more appropriate.

Indicate in the title the name of the model is indeed more appropriate. The title has therefore been changed.

2- I think it is important that the authors nuance a bit more their statements in the introduction about the study of aerosol effects with a regional climate model. While resolution is important, the resolution used is not far from what some ESM models are already using. More importantly, there are other aspects that are key to understand aerosol impacts and at present many ESMs already include aerosol and cloud processes that are far more advanced than the ones represented in this study. The regional model used here includes a simplified aerosol scheme (without aerosol microphysics and number, for example), and many other simplifications (like introducing only the first indirect effect or using a constant nitric acid climatology). Besides that, regional climate modeling cannot account for slow climate responses to aerosols in the domain, and in fact heavily depends upon the ESM driver through the boundaries. I think these aspects and the associated limitations should be clearly explained and discussed both in the introduction and the discussion.

Effectively, the resolution used here is not far from what some ESM models are already using and the representation of the first indirect effect in our model is rather simplified. On the other hand, this simplified representation of the first indirect effect is still used in about 50% of global climate models and the representation of the direct aerosols effect in our model is rather realistic. The limitations of this study (constant nitric acid climatology and simplified indirect aerosol effect) are discussed in the conclusion. The use of a constant nitric acid climatology, which is a major limitation of the study, is now clearly indicated in the text (Page 5, lines 30-34) and in the abstract (Page 1, lines 12-14). Moreover, in this study, the aerosols effects on SST are effectively not taken into account and we therefore focus mainly on fast climate responses to aerosol forcings (Page 4, lines 18-19).

3- Nitric acid is implemented in the model as a constant monthly climatology based on the CAMS reanalysis. One key result reflected in the abstract is the increase in nitrate and its impact upon the DRF. At least, the abstract should clearly acknowledge the constant nitric acid assumption.

This assumption has clearly been added in the abstract (Page 1, lines 12-14). For information, Figure Appendix B1 (will not be shown in the article) presents the $HNO_3$ evolution in 4 CMIP6 models and in the CAMS reanalysis. There is indeed a $HNO_3$ decrease in the future period in the lower layers (925 and 850 hPa). On the other hand, at 750 hPa this decrease is less visible. Moreover, given the differences between the models, the uncertainty remains relatively high. During the nitrate aerosols implementation in the TACTIC aerosol scheme, we also performed a sensitivity test to evaluate the impacts of a time-dependent or a constant nitric acid climatology. Figure Appendix B2 (will not be shown in the article) shows that, in our model, the nitrate concentration is relatively little impacted by the use of a constant or time-dependent nitric acid climatology. For this reason, we decided to use a constant nitric acid climatology in this version of the model (page 5, lines 30-33). Nevertheless, the implementation of a time-dependent nitric acid climatology is envisaged in a future version of the TACTIC aerosol scheme (page 17, lines 5-6).

4- In page 6 you state: "The future period has been selected in the near future because, unlike greenhouse gases, the most important aerosol change is up to the middle of the century. Moreover, the near future horizon period is most suitable to help public decision-makers." I agree with this statement. However, I find that the selection of the reference period (1971-2000) is not well justified and may even been a bit inconsistent with the argument that "is most suitable to help public decision makers". Why the reference period is 1971-2000. Is it because of sulfate and the associated large signal in the DRF? Wouldnât policy makers prefer to see the differences between mid-century and the present day?

Indeed it is not the best choice to help public decision makers but we have chosen this period because it is a reference period commonly used (page 6, line 17).

5- Given that you are using a regional climate model driven by an ESM with a similar

(although not identical) aerosol scheme, they should be compared. In fact, average results of the DRF are compared to other studies throughout the paper but I really miss a clear and consistent comparison with the parent ESM. This could also respond to the question: how useful is resolution and downscaling for the diagnostics that are discussed in the paper? It would be even better if other available CMIP6 ESM results could be consistently compared within exactly the same domain. That would provide a solid comparison reference.

A figure showing the comparison between the RCM and the driving ESM for AOD and surface SW DRF has been added in Appendix (Figure A1 and Figure A2). The forcing model also shows a similar AOD and surface SW DRF trends over Europe and the Mediterranean Sea (page 8, lines 27-29 and page 11, lines 31-34). However, we can observe a surface SW DRF decrease more pronounced with the CNRM-ESM2-1 global model due to the fact that nitrate and ammonium aerosols are not taken into account in this model (page 11, lines 33-34). A comparison with other CMIP6 ESM would indeed be very interesting, but is outside of the scope of this work.

6- Please clearly state that you are calculating the instantaneous RF. Current practice is to calculate effective radiative forcing which includes the forcing and the fast response. Also, why only the direct radiative forcing is discussed in section 3.2 and not the indirect effect? What is the relative role of both upon the total forcing? This is particularly strange as in section 4 you discuss on the interaction with clouds and you suggest influences of the indirect effect upon the certain results.

We have now clearly specified that we calculate the instantaneous RF in section 3.2. (page 10, line 26). Studying the indirect radiative effect is not the main objective of the article because it is not fully taken into account (only the first indirect effect and nitrate particles are not yet included). The primary objective of this study was therefore to focus on the direct radiative forcing of aerosols. On the other hand, the cloud radiative forcing and the forcing due to semi-direct aerosol effects have nevertheless been studied in some interesting regions (central Europe, Iberian peninsula) in a qualitative way

with the different mechanisms involved (sections 4.2 and 4.3). In addition, to complete this study, we have now added a figure presenting the effective radiative forcing (ERF), at the Top Of Atmosphere (TOA), due to aerosol-radiation interactions (ERFari, Figure 13 (d)) and to aerosol-cloud interactions (ERFaci, Figure 14 (b)). These figures are discussed in the text (page 1, lines 22 and 24, page 15, lines 1-4 and page 15 lines 26-27).
* * *
[Figure]

**Fig. 1.** B1 HNO3 evolution in 4 CMIP6 models and in the CAMS reanalysis over the period 1971-2050 at 925, 850 and 700 hPa (not shown in the article)

[Figure]

**Fig. 2.** B2 Nitrate concentration evolution (fine bin on the left and coarse bin in the right) over Europe using a time-dependent (red) or constant (black) hno3 climatology. The blue dotted line was used for a

[Figure]

**Fig. 3.** A1 Total AOD evolution between the past period (1971-2000) and the future period (2021-2050) with the CNRM-ALADIN63 model and its driving ESM.

CNRM-ALADIN63

CNRM-ESM2-1
(driving ESM)

SSP 5-8.5

SSP 3-7.0

SSP 1-1.9

**Fig. 4.** A2 Surface SW DRF evolution between the past period (1971-2000) and the future period (2021-2050) with the CNRM-ALADIN63 model and its driving ESM.

[Figure]

**Fig. 5.** Figure 13 Mean differences, for the months of June, July and August, between SSP585 and SSP585cst simulations over the period 2021-2050 for different parameters

[Figure]

**Fig. 6.** Figure 14 Mean differences, for the months of June, July and August, between SSP585 and SSP585cst simulations over the period 2021-2050 for different parameters